# Consistent Region-Informed Self-supervised Pretraining

## Abstract

Dense prediction tasks such as semantic segmentation require representations that capture both global semantics and local structure. Most self-supervised learning methods prioritise image-level invariance, producing strong features for classification but offering limited guidance for tasks requiring (or depending on) spatial coherence. In parallel, several approaches have been proposed specifically for dense prediction, but their improvements in local fidelity often come at the cost of weaker global transfer. We present CRISP (Consistent Region-Informed Self-Supervised Pretraining), a framework that enhances patch-level learning with explicit region-level alignment. CRISP discovers coherent regions in a reference image, projects them to augmented views via geometric correspondences, and aggregates their patch features into concept tokens with a mask-guided module. By enforcing consistency at the region, patch, and global levels, CRISP learns representations that are both semantically strong and spatially coherent. Pretraining on ImageNet-1K shows that CRISP achieves substantial gains on dense prediction benchmarks while maintaining strong performance on global benchmarks. These results establish region-level consistency as a critical ingredient for advancing universal visual representations.

## 1 Introduction

Self-supervised learning (SSL) has emerged as a powerful paradigm for learning visual representations without manual annotations, often achieving performance on par with, or even surpassing, supervised pretraining in a wide range of downstream tasks. Beyond its strong results on mainstream computer vision problems such as image classification (Ziegler & Asano, 2022), object detection (Amrani et al., 2020), and semantic segmentation (Nandam et al., 2025), SSL has proven particularly valuable in domains where labeled data is scarce, expensive, or impractical to obtain. Examples include medical imaging (Krishnan et al., 2022; Marikkar et al., 2023), where expert annotations require significant time and specialised knowledge; satellite and aerial imagery (Wang et al., 2022), where datasets span vast geographic areas; underwater exploration (Yang et al., 2022), where visibility and conditions make manual labeling challenging; and many others. This broad applicability underscores SSL's potential as a general-purpose representation learning strategy.

Within the broad spectrum of SSL approaches, invariance-based methods have gained particular traction due to their ability to produce robust, transferable features that can be deployed off the shelf without task-specific retraining. An influential direction is the DINO family of methods, starting with DINO (Caron et al., 2021) and its extensions, iBot (Zhou et al., 2022), DINOv2 (Oquab et al., 2024), and DINOv3 (Siméoni et al., 2025), which adopt a teacher–student framework with Vision Transformers (ViTs) to enforce consistency between augmented views. These methods excel at global tasks such as image classification but remain limited on dense prediction tasks. The patch-level objective provides only weak supervision, where the student processes a masked version of an augmented view and the teacher processes the unmasked version of the same view, with alignment enforced only between corresponding patches. This setup ignores cross-view or cross-context relationships, so patches are not encouraged to maintain consistent semantics beyond that single view. In addition, patches are treated as independent units, as if each were a standalone concept, without modeling their spatial coherence or relationships to neighboring patches. As a result, the learned features are globally discriminative but spatially misaligned, which limits their ability to capture fine-grained boundaries and region-level invariances essential for dense tasks.

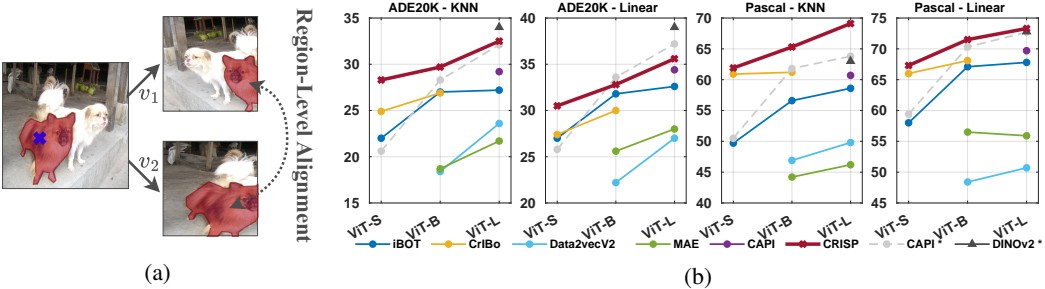

(a)                                                                                 (b)

Figure 1: (a) CRISP discovers a coherent region in the reference image, tracks it across augmented views, consolidates its patch features into a concept token, and aligns student–teacher tokens to enforce region-level consistency. (b) We evaluate frozen representations using $k$-NN and linear probes. CRISP outperforms prior baselines, scales well with model size, and in some cases surpasses models pre-trained on much larger datasets. * pretrained on the LVD-142M dataset ($\approx$142M images).

On the other hand, SSL methods explicitly designed for dense tasks (Xie et al., 2021; Hénaff et al., 2021; Stegmüller et al., 2023; Lebailly et al., 2024; Dukić et al., 2025) emphasise pixel- or region-level supervision, and often rely on heuristics, clustering strategies, or object-centric priors. While effective, these hand-crafted mechanisms can introduce bias and limit scalability beyond the settings they were designed for. Moreover, the strong focus on local detail can dilute global semantics as supervision is concentrated at the patch or region level, models are encouraged to optimise for local consistency rather than building representations that capture object identity, scene context, or cross-image invariances. This in turn weakens transfer on recognition benchmarks. Addressing these issues calls for more principled approaches to dense representation learning, methods that can deliver strong spatial fidelity for dense prediction tasks while also preserving global semantics.

We address these limitations with CRISP (**C**onsistent **R**egion-**I**nformed **S**elf-Supervised **P**retraining), which enforces region-level consistency by leveraging the geometric relationships between augmented views, information that is typically discarded in existing SSL methods. By maintaining explicit links between corresponding regions, CRISP enables their features to be coherently aggregated into higher-level concept tokens. Aligning student–teacher concept tokens then promotes invariance at the region level, complementing global and patch-level objectives and yielding representations that are both semantically rich and spatially precise. Through extensive experiments on ImageNet-1K pretraining, CRISP achieves strong transfer performance across both global and dense benchmarks, with state-of-the-art (SOTA) results on several segmentation and dense prediction benchmarks, refer to Figure 1. Ablation studies further validate the impact of key design choices, showing that these gains come with minimal additional computational cost.

The rest of this paper is organised as follows. We first review related works in Section 2. Section 3 then introduces the CRISP framework, followed by extensive experiments and ablations in Section 4. Finally, Section 5 offers concluding remarks and outlines future directions.

## 2 RELATED WORK

Historically, early SSL methods relied on hand-crafted pretext tasks that encouraged models to learn general-purpose features. Examples include colourisation (Zhang et al., 2016; Larsson et al., 2016; 2017), relative patch location (Doersch et al., 2015), solving jigsaw puzzles (Noroozi & Favaro, 2016; Kim et al., 2018), cross-channel prediction (Zhang et al., 2017), predicting noise (Bojanowski & Joulin, 2017), predicting image rotations (Gidaris et al., 2018), spotting artefacts (Jenni & Favaro, 2018), etc. While these tasks demonstrated the potential of SSL, their objectives were often too narrow or low-level, and the resulting representations did not transfer strongly to downstream tasks.

This led to the emergence of instance discrimination methods, where each image is treated as its own class. (Wu et al., 2018) introduced a memory-bank contrastive method, followed by SimCLR (Chen et al., 2020), which maximise agreement between augmented views of the same image while contrasting against others. While contrastive learning dominated early SSL, later work showed that negative pairs were not strictly necessary. BYOL (Grill et al., 2020) introduced an online–target Siamese framework with a predictor, learning effective features using only positive pairs by relying on

batch normalisation (Ioffe & Szegedy, 2015), while SimSiam (Chen & He, 2021) simplified the setup further by relying on weight sharing and a stop-gradient operation to prevent collapse. Barlow Twins (Zbontar et al., 2021) and VICReg (Bardes et al., 2022) proposed alternative non-contrastive objectives based on redundancy reduction and variance–covariance regularisation, further demonstrating that stable and powerful representations can be learned without negatives. Clustering-based SSL offered another direction. DeepCluster (Caron et al., 2018) and SwAV (Caron et al., 2020) performed online clustering of features and used cluster assignments as pseudo-labels, thereby avoiding trivial collapse without explicit negatives.

The introduction of Vision Transformers (ViTs) enabled richer structure. DINO (Caron et al., 2021) applied teacher–student self-distillation to ViTs and showed that attention maps could localise objects without supervision, revealing implicit object sensitivity. Building on this, iBOT (Zhou et al., 2022) combined masked image modeling with distillation by predicting teacher patch tokens, leading to semantically meaningful patch features and better transfer to dense tasks. DINOv2 (Oquab et al., 2024) scaled this recipe to hundreds of millions of curated images and billion-parameter ViTs, producing universal features transferable across recognition, segmentation, and depth estimation. DINOv3 (Siméoni et al., 2025) further scaled to multi-billion parameter models with training refinements like Gram anchoring, yielding dense representations competitive with supervised pretraining.

In parallel, masked image modeling (MIM) (He et al., 2022; Xie et al., 2022; Atito et al., 2023) emerged as a complementary paradigm, training models to reconstruct randomly masked patches with an asymmetric encoder–decoder. MIM excels at learning spatially grounded, fine-grained features but often struggles to capture high-level semantics, leading to weaker transfer on discriminative tasks.

Despite these advances, a gap remains. Invariance-based SSL produces strong global semantics but lacks spatial precision, while reconstruction- or patch-based SSL captures local detail yet weakens semantic coherence. This tension has motivated dense-focused SSL. Early approaches incorporated region-level consistency using external cues, where some methods rely on pretrained models and offline correspondence discovery to match regions across views, while others generate pseudo-segmentation labels via heuristics before training (Xie et al., 2021; Hénaff et al., 2021). Building on these ideas, CrOC (Stegmüller et al., 2023) learns dense region representations by jointly clustering patch embeddings across two augmented views, enabling unsupervised segmentation without external proposals. CrIBo (Lebailly et al., 2024) takes this further by aligning object-level nearest neighbors across different images to disentangle multiple objects within scene-centric datasets. OCEBO (Dukić et al., 2025) instead targets object-centric slot architectures, bootstrapping a target encoder via self-distillation and filtering uninformative patches to achieve unsupervised object discovery on real-world data. While effective, these approaches often depend on heuristics, clustering, or object-centric priors, and can trade off global semantics for local fidelity.

Our work, CRISP, closes this gap by enforcing region-level consistency across augmented views without relying on external labels or heuristics. In doing so, it learns representations that are both globally semantic and spatially precise, effectively unifying the strengths of global and dense SSL.

## 3 METHODOLOGY

CRISP complements invariance-based SSL with the spatial precision needed for dense prediction. We build on iBOT (Zhou et al., 2022), selected for its strong performance and architectural simplicity. At its core, iBOT combines a global image-level loss on the [CLS] token with a patch-level loss that aligns masked and unmasked patch embeddings within a single view. While effective for image-level tasks, this design treats patches as independent units, limiting spatial precision and region-level invariance. CRISP addresses this by adding a region-consistency objective that enforces alignment of coherent regions across views. The framework unfolds in three stages: (1) region discovery via similarity maps (Sec. 3.1), (2) cross-view region warping using geometric transforms (Sec. 3.2), and (3) mask-guided concept token aggregation (Sec. 3.3). An overview of CRISP is shown in Figure 2.

### 3.1 REGION DISCOVERY

The first step in CRISP is to identify coherent regions within a reference image in a fully self-supervised manner. While ViTs naturally operate on fixed-size patches, meaningful visual concepts rarely align with the boundary of a single patch. Instead, semantically consistent parts of a scene,

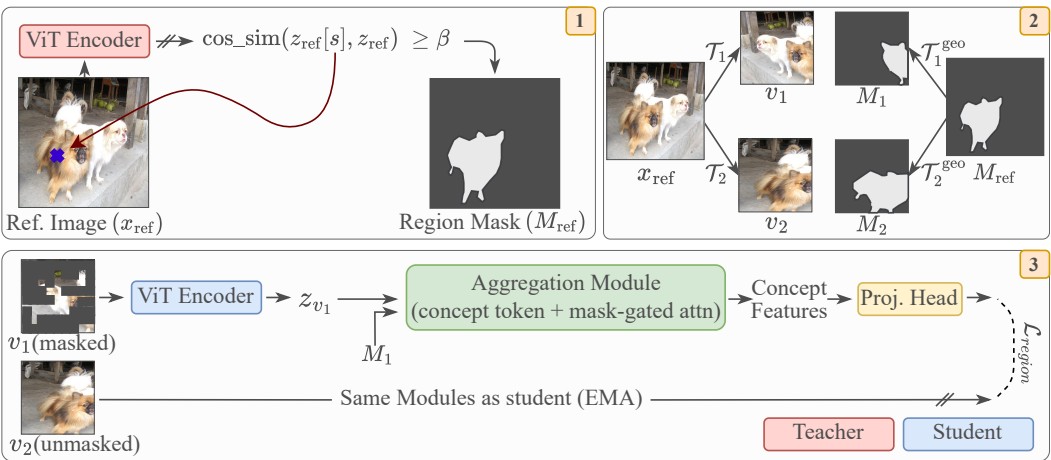

Figure 2: Overview of CRISP. (1) Region discovery: a seed patch computes cosine similarity to all patches from teacher embeddings; thresholding yields a coherent region mask. (2) Cross-view warping: the mask is mapped to augmented views via geometric transforms. (3) Mask-guided aggregation: region patches are consolidated into a concept token for student–teacher matching.

such as head of an animal, door of a house, or handle of a cup, emerge as groups of correlated patches. Our goal is therefore to discover such regions directly from self-supervised features.

To capture such regions, we leverage the semantic structure already present in the iBOT teacher network, which we use as initialisation and continue to update during CRISP training. Its patch embeddings provide a strong starting point for region discovery, allowing us to form contiguous, semantically coherent areas that evolve as training progresses.

Formally, let $x \in \mathbb{R}^{3 \times H_0 \times W_0}$ denote the original image. To standardise the patch grid across the batch, we resize $x$ to a fixed square resolution $H \times W$, yielding the *reference image* $x_{\mathrm{ref}}$. This image is partitioned into $n \times n$ non-overlapping patches of size $p \times p$ pixels, where $p = H/n$.

The reference image is passed through the teacher ViT, and patch embeddings are extracted by averaging the outputs of the last $M$ transformer blocks to stabilise the similarity maps, obtaining $z_{\mathrm{ref}} \in \mathbb{R}^{n^2 \times d}$, where $d$ is the embedding dimension of the ViT. Averaging the highest layers reduces block-level noise and captures the stronger semantics present near the network head, yielding cleaner, more contiguous region masks.

We then uniformly sample a seed patch token index $s \in \{1, \ldots, n^2\}$ and compute the cosine similarity between the seed and all other patch tokens in the image:

$$a_s(i) = \frac{\langle z_{\mathrm{ref}}[s],\ z_{\mathrm{ref}}[i] \rangle}{\|z_{\mathrm{ref}}[s]\|_2 \cdot \|z_{\mathrm{ref}}[i]\|_2}, \quad i = 1, \ldots, n^2 \tag{1}$$

where $\langle \cdot, \cdot \rangle$ denotes the dot product and the denominator applies $\ell_2$ normalisation. Finally, we reshape the $a_s$ into $n \times n$, producing a spatial similarity map, which is then thresholded:

$$M_{\mathrm{ref}}(i, j) = \mathbb{1}\big[a_s(i, j) \geq \beta\big], \quad M_{\mathrm{ref}} \in \{0, 1\}^{n \times n}, \tag{2}$$

where $\beta$ is a fixed similarity threshold. Because the teacher is initialised from iBOT and continues to evolve under CRISP training, its embeddings reliably produce coherent, meaningful regions. The region may not match exact object boundaries, but it is sufficient for reliable region-level alignment.

## 3.2 CROSS-VIEW REGION WARPING

Once a region is identified in the reference image, the next step is to track it across augmented views $v_1$ and $v_2$. Each view is generated from the original image $x$ via a composition of geometric and photometric transformations: $\mathcal{T}_k = \mathcal{T}_k^{\mathrm{geo}} \circ \mathcal{T}_k^{\mathrm{photo}}, \quad k \in \{1, 2\}$, where $\mathcal{T}_k^{\mathrm{geo}}$ includes operations such as random cropping, resizing, and flipping, while $\mathcal{T}_k^{\mathrm{photo}}$ applies only appearance changes such as color jittering, blurring, or grayscale conversion, that do not alter spatial coordinates.

Since $M_{\text{ref}}$ is defined in patch coordinates on the resized reference image ($H \times W$), we first convert its active patch indices into pixel coordinates in the original image resolution ($H_0 \times W_0$). This produces a set of pixel-coordinate regions corresponding to the discovered coherent region in the original image. We then apply the recorded geometric transformation $\mathcal{T}_k^{\text{geo}}$ directly to these coordinates, yielding the region location in view $v_k$ without any intermediate mask resampling. The transformed coordinates are finally discretised into the patch grid of $v_k$, producing $M_k \in \{0,1\}^{\hat{n}_k \times \hat{n}_k}, \quad k \in \{1,2\}$.

By working directly in coordinate space rather than resampling masks after each augmentation, this procedure preserves exact geometric alignment between $M_1$ and $M_2$, guaranteeing that they correspond to the same region in both augmented views. At the same time, it avoids costly resampling operations, making the method efficient and naturally suited for parallel execution across a batch.

### 3.3 Mask-Guided Concept Token Aggregation

Once a region is localised in both views, CRISP aggregates its information into a single concept representation. Let $z_k \in \mathbb{R}^{\hat{n}_k^2 \times d}$ denote the last-block ViT patch embeddings of augmented view $v_k$ (student if $k = 1$, teacher if $k = 2$). From Sec. 3.2, the warped mask $M_k$ defines the region index set $\mathcal{I}_k = \{ i \in \{1, \ldots, \hat{n}_k^2\} \; : \; M_k(i) = 1 \}$. We then prepend a learnable concept token $c_k^{(0)} \in \mathbb{R}^d$ to the sequence of patch embeddings, yielding

$$Z_k^{(0)} = \left[ c_k^{(0)} ; \; z_k(1), z_k(2), \ldots, z_k(\hat{n}_k^2) \right] \; \in \; \mathbb{R}^{(1+\hat{n}_k^2) \times d}. \tag{3}$$

To restrict the aggregation to the discovered region, we construct an additive mask $\Pi_k \in \mathbb{R}^{(1+\hat{n}_k^2) \times (1+\hat{n}_k^2)}$ such that only the concept token (index 0) can attend to the region patches:

$$\Pi_k(u, v) = \begin{cases} 0, & u = 0 \;\; \text{and} \;\; v \in \mathcal{I}_k, \\ -\infty, & \text{otherwise.} \end{cases} \tag{4}$$

In practice, $-\infty$ is implemented as a large negative constant. This ensures that the concept token aggregates information exclusively from the region, and no other token interactions are permitted.

For a transformer block $\ell = 1, \ldots, L$, with $Q^{(\ell)}, K^{(\ell)}, V^{(\ell)}$ the standard query, key, and value projections of $X_k^{(\ell-1)}$, masked attention is computed as

$$\text{Attn}(Q^{(\ell)}, K^{(\ell)}, V^{(\ell)}; \Pi_k) \; = \; \text{softmax}\left( \frac{Q^{(\ell)} K^{(\ell)\top}}{\sqrt{d}} + \Pi_k \right) V^{(\ell)}. \tag{5}$$

After $L$ such layers (with residual and MLP sublayers), the updated concept token is taken as the region embedding $c_k \in \mathbb{R}^d$.

### 3.4 Overall Training Objective

CRISP extends iBOT by adding a region-level objective. The iBOT losses, $\mathcal{L}_{\text{global}}$ and $\mathcal{L}_{\text{patch}}$, follow a teacher–student setup with online centering and temperature scaling. For the region objective, each region embedding $c_k$ is projected as $h_k = g_k(c_k) \in \mathbb{R}^{d'}$, where $g_1$ and $g_2$ are two-layer MLPs with normalisation for student and teacher. Only the concept token is aligned where the student $h_1$ is matched with the teacher $h_2$ through a region-consistency loss, ensuring stable representations across views. To ensure that regions correspond to meaningful concepts, we exclude cases where either augmented view contains fewer than $t$ patches. This prevents matching regions that are too small to capture semantic structure, or are not visible in one of the views. In our experiments, we set $t = 4$.

$$\mathcal{L}_{\text{region}} \; = \; \frac{1}{2}\left[ \ell_{\text{reg}}(h_1, \text{sg}(h_2)) + \ell_{\text{reg}}(h_2, \text{sg}(h_1)) \right], \tag{6}$$

where $\text{sg}(\cdot)$ denotes stop-gradient and $\ell_{\text{reg}}$ is the same softmax cross-entropy over a teacher queue used in the global and patch objectives.[1] The full CRISP loss combines all objectives:

$$\mathcal{L}_{\text{CRISP}} \; = \; \mathcal{L}_{\text{global}} \; + \; \lambda_{\text{patch}} \mathcal{L}_{\text{patch}} \; + \; \lambda_{\text{region}} \mathcal{L}_{\text{region}}, \tag{7}$$

with $\lambda_{\text{patch}}$ and $\lambda_{\text{region}}$ are set to 1.0 for simplicity.

This multi-scale design integrates complementary signals where global objectives capture semantics, patch objectives enforce local precision, and region-consistency bridges the two.

---

[1]Other contrastive or distillation-based losses could be used here; we follow iBOT for consistency.

# 4 EXPERIMENTS

We assess the effectiveness of CRISP framework through a comprehensive set of experiments. Sec. 4.1 details the experimental setup. Sec.4.2 presents quantitative and qualitative results across both dense prediction tasks and whole-image understanding tasks. Visualisaions are shown in Sec. 4.3 Finally, Sec. 4.4 provides targeted ablations that systematically validate the design choices of CRISP.

## 4.1 EXPERIMENTAL SETUP

We pretrain our models on ImageNet-1K without labels, using ViTs (ViT-S/16, ViT-B/16, and ViT-L/16) as backbones with a patch size of $16 \times 16$. For region discovery, we average teacher features from the last $M = 4$ blocks, pick a seed patch uniformly at random, and threshold the resulting similarity map with $\beta = 0.75$. For concept modeling, we add $L = 1$ transformer block whose hidden size matches the backbone (e.g., 384 for ViT-S). The concept token and the patch tokens share a projection head with two linear layers of width 2048 and GELU activations, followed by a 256-dimensional bottleneck. The CLS token uses the same design. All outputs are $\ell_2$ normalised and each token type (concept, CLS, patch) is mapped to an 8192-dimensional embedding with its own final linear layer. We follow the iBOT pretraining recipe, using AdamW with a cosine learning-rate schedule. Training runs on four GPUs with per-GPU batch sizes of 64 for ViT-S, 40 for ViT-B, and 32 for ViT-L, and all models are pretrained for 200 epochs from iBOT initialisation.

## 4.2 MAIN RESULTS

**Semantic Segmentation.** A key aim of CRISP is to learn semantically rich, spatially localised representations from patch features. We evaluate them using the $k$-NN and linear protocols of CAPI (Darcet et al., 2025), reporting mean Intersection-over-Union (mIoU) on ADE20K (Zhou et al., 2017), PASCAL VOC (Everingham et al., 2010), and Cityscapes (Cordts et al., 2016); refer to Table 1. CRISP consistently outperforms prior approaches, most notably surpassing CAPI by a large margin, particularly with small encoders, despite CAPI being pre-trained on $100\times$ more data, and performs on par with DINOv2. Across datasets, our $k$-NN scores show the largest gains, which emphasises the strength and transferability of CRISP's patch-level embeddings without task-specific heads.

Table 1: Comparison with SOTA methods on semantic segmentation using frozen features. We report $k$-NN and linear probe performance. All models are evaluated with input resolution adapted to 256 patch tokens (i.e. $224 \times 224$ for patch size 14, $256 \times 256$ for patch size 16).

| Model | ADE-20K | | Pascal-VOC | | Cityscapes | |
|---|---|---|---|---|---|---|
| | $k$-NN | Linear | $k$-NN | Linear | $k$-NN | Linear |
| | ViT-Small | | | | | |
| SiT | 17.6 | 22.3 | 37.6 | 47.1 | 31.4 | 35.6 |
| Dino | 17.5 | 20.9 | 37.2 | 42.3 | 30.9 | 35.4 |
| iBot | 22.0 | 27.0 | 49.7 | 58.0 | 33.5 | 37.9 |
| CrOC | 19.6 | 24.2 | 49.0 | 58.5 | 29.1 | 34.6 |
| CrIBo | 24.9 | 27.4 | 60.9 | 66.0 | 33.3 | 37.9 |
| CRISP (Ours) | **28.2** | **30.5** | **61.9** | **67.3** | **34.3** | **39.1** |
| CAPI* | 20.6 | 25.8 | 50.5 | 59.4 | 31.3 | 36.5 |
| | ViT-Base | | | | | |
| MAE | 18.7 | 25.6 | 44.2 | 56.5 | 32.2 | 38.1 |
| Data2Vec 2.0 | 18.4 | 22.2 | 46.9 | 48.4 | 28.3 | 35.0 |
| iBot | 27.0 | 31.8 | 56.6 | 67.1 | 35.9 | 39.6 |
| CrIBo | 26.9 | 30.0 | 61.2 | 68.1 | 33.7 | 38.4 |
| CRISP (Ours) | **29.7** | **32.8** | **65.3** | **71.5** | **36.0** | **41.0** |
| CAPI* | 28.3 | 33.6 | 61.8 | 70.3 | 36.9 | 41.9 |
| | ViT-Large | | | | | |
| MAE | 21.7 | 28.0 | 46.2 | 55.9 | 34.9 | 39.6 |
| Data2Vec 2.0 | 23.6 | 27.0 | 49.8 | 50.7 | 33.1 | 38.6 |
| iBot | 27.2 | 32.6 | 58.6 | 67.8 | 36.7 | 41.9 |
| CAPI | 29.2 | 34.4 | 60.7 | 69.7 | 35.6 | 41.7 |
| CRISP (Ours) | **32.5** | **35.6** | **69.1** | **73.3** | **36.9** | **42.2** |
| DINOv2* | 34.0 | 39.0 | 63.0 | 72.8 | 42.0 | 46.8 |
| CAPI* | 32.1 | 37.2 | 63.8 | 72.7 | 38.9 | 44.3 |

**Video Instance Segmentation.** Table 2 evaluates frozen features for mask propagation on DAVIS-2017 (Caelles et al., 2018), YouTube-VOS (Xu et al., 2018), and MOSE (Ding et al., 2023). Following the nearest-neighbour protocol of DINO, we propagate masks frame-to-frame using cosine similarity in feature space, without any fine-tuning. Grey rows list larger backbones for reference.

Under identical backbones and pretraining data, CRISP surpasses SOTA methods, while remaining competitive with much larger models, showing that its region-consistent pretraining produces stable, object-aligned features transferable to video tracking without task-specific training.

Table 2: Video segmentation tracking evaluation. We report mean region similarity ($\mathcal{J}_m$), mean contour accuracy ($\mathcal{F}_m$), and their average ($\mathcal{J}\&\mathcal{F}$). All videos are resized to 480p.

| Method | Model | DAVIS | | | YouTube-VOS | | | MOSE | | |
|---|---|---|---|---|---|---|---|---|---|---|
| | | $\mathcal{J}\&\mathcal{F}$ | $\mathcal{J}_m$ | $\mathcal{F}_m$ | $\mathcal{J}\&\mathcal{F}$ | $\mathcal{J}_m$ | $\mathcal{F}_m$ | $\mathcal{J}\&\mathcal{F}$ | $\mathcal{J}_m$ | $\mathcal{F}_m$ |
| iBot | ViT-S/16 | 61.8 | 60.4 | 63.2 | 66.8 | 66.1 | 67.4 | **38.0** | 34.1 | 41.6 |
| CrOC | ViT-S/16 | 60.2 | 58.8 | 61.5 | 64.3 | 63.9 | 64.7 | 34.6 | 31.1 | 38.1 |
| CrIBo | ViT-S/16 | 61.5 | 60.0 | 63.0 | 66.5 | 65.9 | 67.2 | 36.7 | 33.1 | 40.4 |
| CRISP | ViT-S/16 | **63.8** | **62.2** | **65.4** | **67.1** | **66.4** | **67.8** | 37.8 | **34.1** | **41.6** |
| DINOv2* | ViT-g/14 | 63.9 | – | – | 65.6 | – | – | 40.4 | – | – |
| Web-DINO (Fan et al., 2025) | ViT-7B/14 | 57.2 | – | – | 43.9 | – | – | 24.9 | – | – |

**Multi-label Classification.** We evaluate linear probing for multi-label recognition in both low-shot and full-data regimes using a ViT-S/16 encoder on $224 \times 224$ resolution. On PASCAL VOC (Everingham et al., 2010), we create 1-, 2-, and 5-shot splits by randomly sampling images per class with a fixed seed. A linear classifier is trained on frozen features, and mAP is reported on the full validation set. Full-set results are further reported on PASCAL VOC, MS COCO (Lin et al., 2014), and Visual Genome (Krishna et al., 2017). Features are formed by concatenating the [CLS] token with averaged patch features. Results are shown in Table 3.

**Multi-class Classification.** We also evaluate CRISP under standard $k$-NN and linear probing.

Across both settings, CRISP demonstrates strong low-shot performance, highlighting its data efficiency. In contrast, dense-task methods such as CROC and CRIBO are decent on dense prediction but underperform severely on classification, underscoring CRISP's versatility and balanced design.

Table 3: mAP results for low-shot multi-label classification on PASCAL VOC as long with the performance on MSCOCO (MC) and Visual Genome (VG)

| | Pascal VOC | | | | MC | VG |
|---|---|---|---|---|---|---|
| | 1 Img | 2 Imgs | 5 Imgs | Full | | |
| iBoT | 44.3 | 58.8 | 70.2 | 93.9 | 58.2 | 32.3 |
| CrOC | 48.7 | 56.9 | 67.8 | 89.4 | 56.2 | 28.8 |
| CrIBo | 48.8 | 60.4 | 70.0 | 92.9 | 59.9 | 32.0 |
| CRISP | **49.3** | **61.0** | **71.1** | **95.2** | **63.0** | **33.0** |

Table 4: $k$-NN and Linear results on multi-class classification benchmarks.

| | $k$-NN | | | Linear |
|---|---|---|---|---|
| | 1% | 10% | 100% | |
| iBoT | 62.3 | 68.9 | 75.1 | 77.9 |
| CrOC | 49.0 | 58.0 | 66.5 | 71.4 |
| CrIBo | 58.1 | 64.4 | 70.7 | 74.9 |
| CRISP | **63.7** | **69.4** | **75.2** | **78.0** |
| CAPI* | – | – | – | 71.5 |

## 4.3 VISUALISATIONS

**Sparse Correspondence.** We evaluate patch matching between two images of the same class, following iBOT's strategy. Using a ViT-L/16 pre-trained on ImageNet-1K with CRISP, we visualise the top 12 correspondences by self-attention on ImageNet validation pairs. As shown in Figure 3, CRISP establishes meaningful matches despite large variations in texture, color, pose, and background, highlighting the robustness and generalisation of its representations. Additional examples between augmented views and across images from same class are provided in the Appendix.

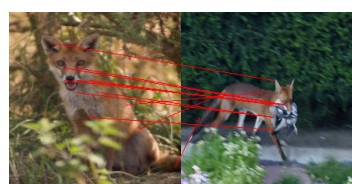 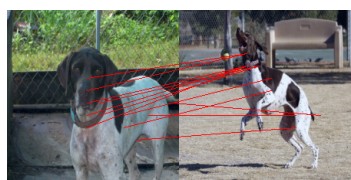 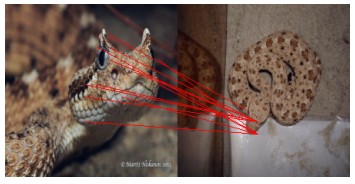

Figure 3: Top 12 patch correspondences between two different images from same class with ViT-L/16.

**Dense Feature Representations via PCA.** Following CAPI (Darcet et al., 2025), we perform PCA on dense output features for qualitative evaluation. Figure 4 visualises the first three principal components as an RGB composite, comparing CRISP against DINOv2 (trained on LVD 142M samples), employing ViT-L backbone. CRISP produces more discriminative and spatially coherent feature maps, where object boundaries emerge clearly with minimal noise. Additional baseline comparisons are presented in the Appendix, along with visualisations of the first six principal components individually, which further illustrate their distinct spatial structures.

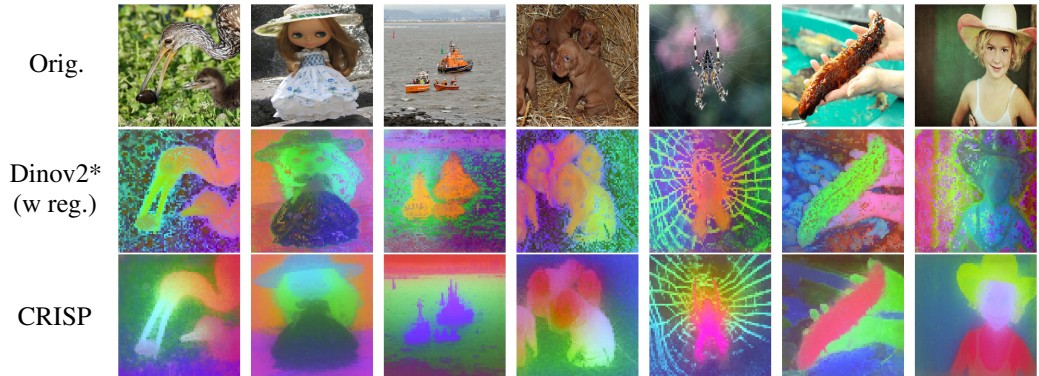

Figure 4: PCA visualisations of dense features, showing CRISP yields clearer, more coherent object boundaries than DINOv2 (ViT-L, LVD 142M).

**Discovered Regions.** Figure 5 shows regions discovered after pretraining. Additional visualisations of discovered regions at the beginning and end of training are provided in the Appendix.

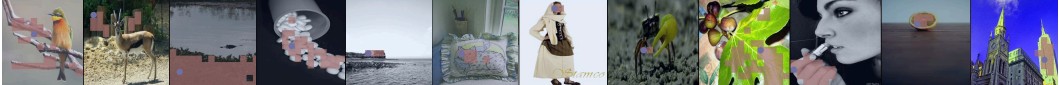

Figure 5: The blue dot marks the seed; the red area shows its discovered region.

## 4.4 ABLATION STUDIES

To better understand the contribution of each component in CRISP, we conduct a series of controlled ablation studies. All ablations use the same ImageNet-1K pretraining protocol with a ViT-S backbone trained for 50 epochs on 4 GPUs with a batch size of 64, starting from iBOT initialisation. We evaluate both global representation quality (10% ImageNet-1K k-NN) and spatial coherence for dense tasks (PASCAL VOC k-NN and linear). The default design choices of CRISP are highlighted in grey.

**Layer Choice for Region Discovery** ($M$). We compare using features from the last layer only versus averaging the last 2, 4, or 6 layers for region discovery. Averaging stabilises similarity maps by reducing block-level noise. As shown in Table 5, the best trade-off arises from averaging the last four layers, while six layers overly smooth features and blur boundaries.

**Effect of $\beta$ on Region Discovery.** Region masks are built by thresholding cosine similarities between a seed patch and all others, with $\beta$ varied from 0.4 to 0.99. Low thresholds capture larger regions that aid global features but add noise, weakening dense-task accuracy (though still above baseline). High thresholds produce very compact regions, limiting their effectiveness for dense tasks. As shown in Figure 6, we find $\beta = 0.75$ to be the best trade-off, yielding stable, coherent masks that align well across views and support both global and dense objectives. At the extreme, $\beta = 0.99$ collapses regions to essentially the same patch across views; while this surprisingly improves over baseline, the lack of region-level context injects noise and reduces global performance.

**Number of Transformer Blocks in the Mask Guided Aggregation Module** ($L$). The module uses a lightweight transformer to merge region tokens into a single concept embedding. As a baseline ($L$=0), we replace the transformer with simple averaging before passing the features to the projection head. We then vary the number of blocks $L \in 1, 2, 4, 6$. As shown in Table 6, gains diminish beyond

$L = 2$. Although $L = 2$ achieves the highest accuracy, $L = 1$ provides the best balance of accuracy and efficiency. Limited improvements at higher $L$ may stem from additional randomly initialissed parameters, which could become more effective with further optimisation.

**Projection Head Sharing.** CRISP uses separate MLP heads for [CLS], patch, and region tokens. We test shared vs. partially shared heads and find full separation is crucial as showing Table 7, [CLS] captures global objectives, while patch/region encode spatial detail.

**Normalisation Sharing.** We test sharing the final normalisation layers between [CLS] and patch tokens. Sharing harms dense transfer since they follow different distributions, while separate normalisations preserve specialisation and improve dense performance without affecting classification.

Table 5: Layer Choice for Region Discovery.

| $M$ | Pascal VOC | | INet-10% |
| | k-NN | Linear | k-NN |
|---|---|---|---|
| 1 | 55.6 | 64.4 | 69.0 |
| 2 | 56.3 | 64.4 | 69.0 |
| 4 | 57.6 | 64.7 | 69.1 |
| 6 | 56.4 | 64.4 | 68.8 |

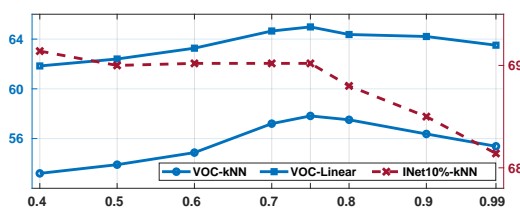

Figure 6: Effect of $\beta$ on Region Discovery.

Table 6: Number of Transformer blocks in the Mask-guided Aggregation Module.

| $L$ | Pascal VOC | | INet-10% |
| | k-NN | Linear | k-NN |
|---|---|---|---|
| 0 | 54.6 | 63.2 | 69.0 |
| 1 | 57.4 | 64.7 | 69.1 |
| 2 | 57.6 | 64.7 | 69.1 |
| 4 | 56.4 | 64.4 | 68.8 |
| 6 | 56.1 | 64.3 | 68.8 |

Table 7: Projection head ablations. Notation: ✓=all shared; ✗=all separate; C∣PR=cls separate, patch and region shared.

| MLP | Last Layer | Pascal VOC | | INet- 10% |
| | | k-NN | Linear | k-NN |
|---|---|---|---|---|
| ✓ | ✓ | 55.6 | 63.9 | 67.9 |
| ✓ | ✗ | 56.3 | 64.4 | 68.9 |
| ✓ | C∣PR | 56.3 | 64.0 | 69.0 |
| ✗ | ✗ | 56.7 | 64.9 | 69.0 |
| C∣PR | C∣PR | 57.0 | 65.0 | 69.0 |

# 5 DISCUSSION AND CONCLUDING REMARKS

We introduced CRISP, a framework that explicitly enforces region-level consistency across augmented views while preserving the strengths of global and patch-level objectives. By discovering coherent regions, warping them across views, and aggregating their features into concept tokens, CRISP achieves high performance on dense tasks and bridges the gap between globally discriminative but spatially coarse representations. CRISP demonstrates strong and versatile representations on ImageNet-1K, improving dense tasks such as segmentation and video tracking while remaining competitive on classification. Remarkably, it achieves these results with modest resources, highlighting its efficiency. Visualizations further show clear object boundaries and robust correspondences, underscoring its interpretability beyond standard invariance-based methods.

**Limitations and Future Work.** While CRISP performs strongly on both dense and global tasks, several limitations remain. Our region discovery relies on a fixed similarity threshold applied uniformly across images, which may not adapt to scene complexity. Future work could explore adaptive or learnable thresholds, or gradually relax them during training for greater robustness. CRISP also treats regions as sets of correlated patches without explicitly modeling spatial connectivity. Introducing constraints such as radius-based neighborhoods or topology-aware grouping may yield more coherent regions. Moreover, our experiments were conducted with modest resources; scaling model size, training duration, and hyperparameter tuning could unlock further gains. Finally, extending CRISP to domains such as medical imaging, remote sensing, or video analysis, potentially with task-aware augmentations and multimodal inputs, offers exciting directions.

**Reproducibility Statement:** We ensure reproducibility by detailing all pretraining experimental setup in Sec. 4.1, evaluation protocols in the Appendix, and ablations in Sec. 4.4. Complete code and configurations are included in the supplementary material and will be publicly released upon acceptance along with the pretrained weights to enable direct replication and extension of CRISP.

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

# A APPENDIX

## A.1 EXPERIMENTAL SETUP FOR EVALUATION

All models are trained and evaluated under consistent conditions. For semantic segmentation, we follow the official evaluation code provided by CAPI without modification, using a single GPU. All models are evaluated at an input resolution corresponding to 256 patch tokens (i.e., $224 \times 224$ for patch size 14 and $256 \times 256$ for patch size 16). For video instance segmentation, we adopt the evaluation code from DINO. Features are extracted from the last four transformer blocks, averaged, and then used for evaluation. The input resolution is set to $480 \times 480$, and training is conducted on a single GPU. For multi-label and multi-class classification, we employ a linear probing setup using 4 GPUs. Each model is trained with inputs of size $224 \times 224$, a learning rate of 0.001, and a per-GPU batch size of 256. We train Pascal VOC for 500 epochs, while MS-COCO, Visual Genome, and linear evaluations on ImageNet-1K are trained for 200 epochs.

## A.2 VISUALISATIONS

**Sparse Correspondence.** We evaluate CRISP on a sparse correspondence task where patches from two images of the same semantic class are expected to match. Our approach follows the strategy introduced by iBOT. To assess performance, we visualise the top 12 correspondences with the highest self-attention scores, obtained from a ViT-L/16 model pretrained on ImageNet-1K with CRISP. The image pairs are sampled from the ImageNet validation set.

Figure 7 and Figure 8 show representative examples. In Figure 7, CRISP achieves near-perfect matching between two augmented views of the same image, accurately aligning almost all patch pairs. In Figure 8, CRISP establishes meaningful correspondences across different images of the same class, despite large variations in texture, color, pose, and background. These results highlight the robustness and generalisation of CRISP's learned representations and demonstrate their suitability for fine-grained, patch-level retrieval tasks.

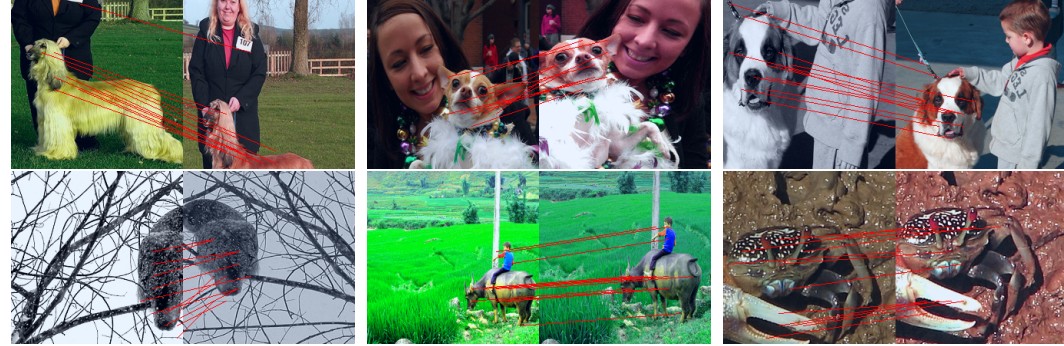

Figure 7: Top 12 patch correspondences between two augmented views from same image

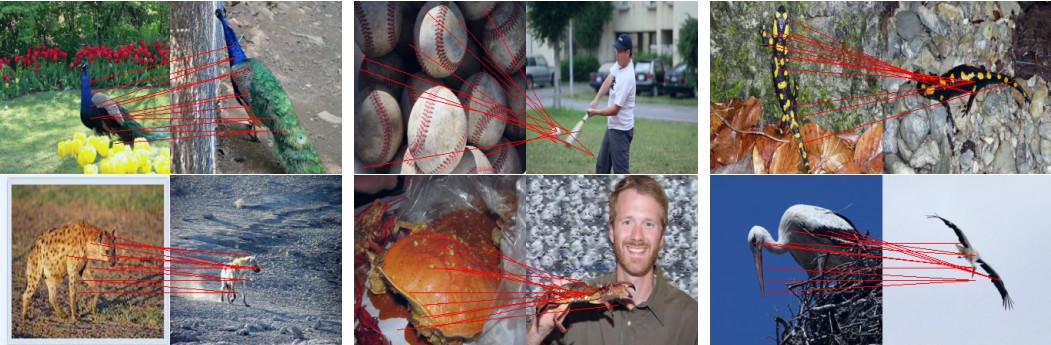

Figure 8: Top 12 patch correspondences across two different images from the same class.

**Qualitative Comparison of Dense Feature Representations via PCA.** Following the qualitative feature analysis proposed in CAPI (Darcet et al., 2025), we apply PCA to the dense output features. In Figure 9, the first three principal components are visualised as an RGB composite, comparing CRISP with state-of-the-art vision models using the ViT-L backbone (except I-JEPA, which employs ViT-H).

Across methods, CRISP produces some of the most discriminative and spatially coherent feature maps. The visualizations clearly delineate object boundaries with minimal noise in uniform regions. Compared to CAPI and DINOv2, CRISP yields cleaner features, while in contrast to masked image modeling (MIM) methods, it focuses more effectively on semantically meaningful regions.

Figure 10 provides a breakdown of the features. The second column again shows the first three principal components as an RGB composite, while the following six columns depict each of the first six components individually. Each component captures distinct semantic regions, highlighting CRISP's ability to encode meaningful visual concepts. Notably, components separate object parts from background clutter, demonstrating the model's capacity to disentangle structured elements of the scene. These results underscore the spatially localised and interpretable nature of CRISP's learned representations.

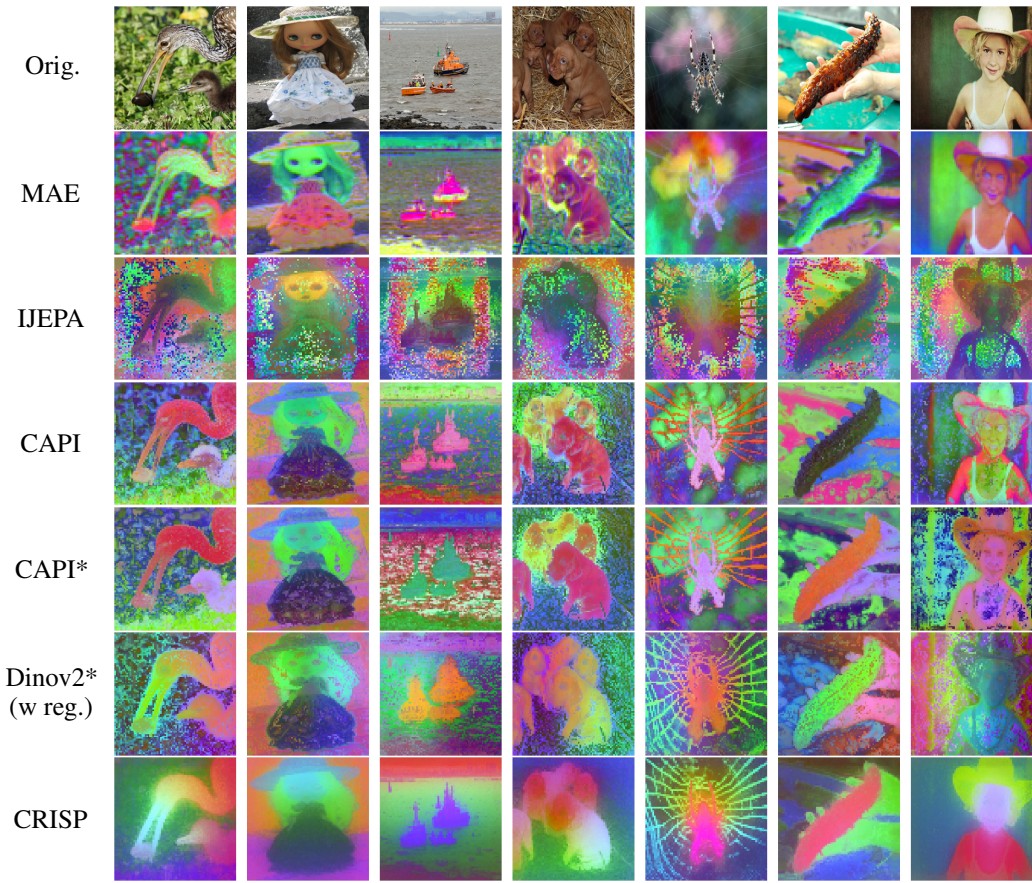

Figure 9: Comparison of dense features. We compare several vision backbones by projecting their dense outputs using PCA and mapping them to RGB.

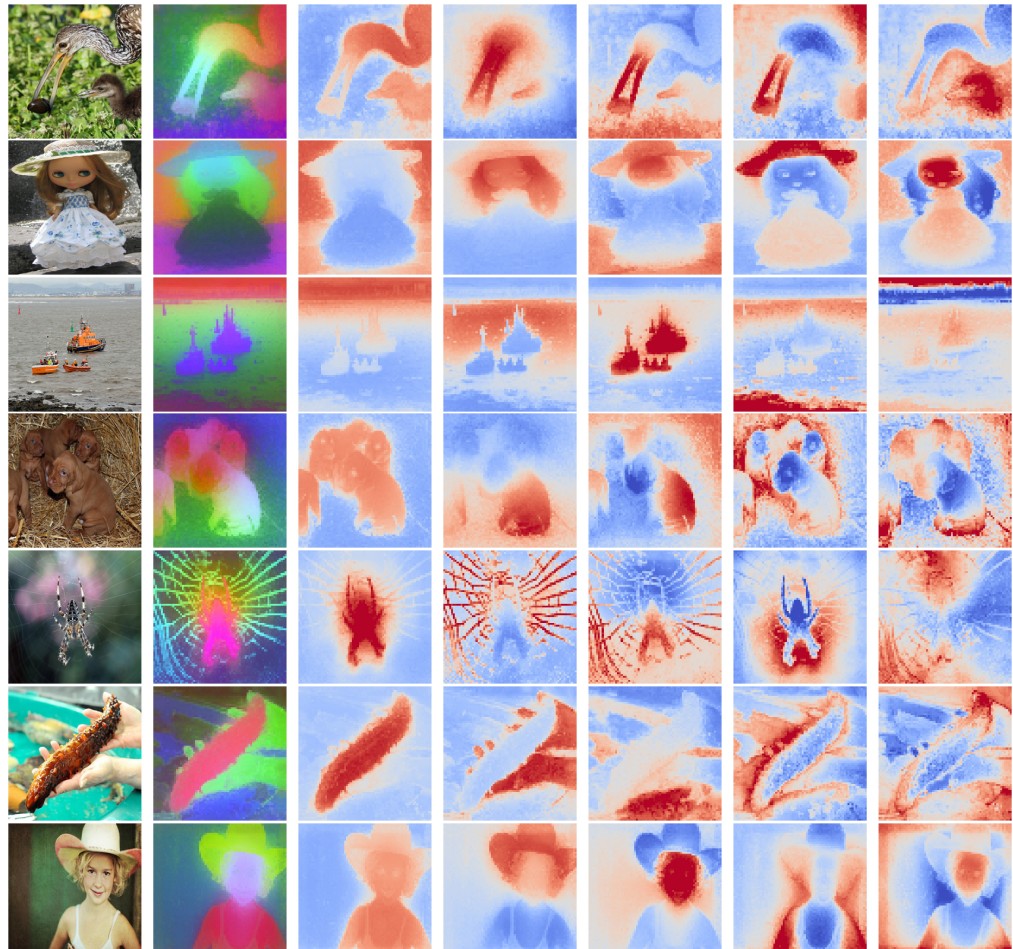

Figure 10: Visualization of features produced by CRISP with a ViT-L/16 model on images at 1120-pixel resolution. The images are randomly sampled from the ImageNet-1K validation set.

**Performance vs. Epochs.** We analyse how performance evolves with pretraining in Figure 11, reporting results on ADE20K, PASCAL VOC, and Cityscapes. We observe a slight performance dip in the early stage, likely due to the introduction of new parameters, after which the model quickly recovers and continues to improve steadily with more pretraining. This trend highlights the stability of CRISP and its ability to benefit consistently from extended pretraining.

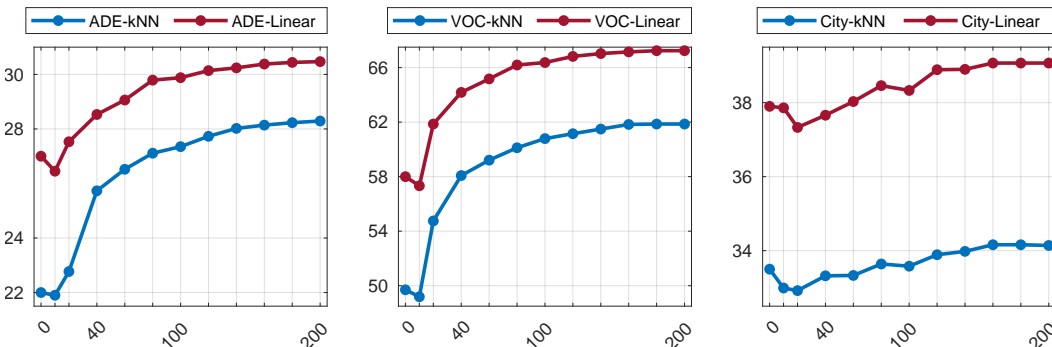

Figure 11: Performance on ADE20K, PASCAL VOC, and Cityscapes as a function of pretraining epochs

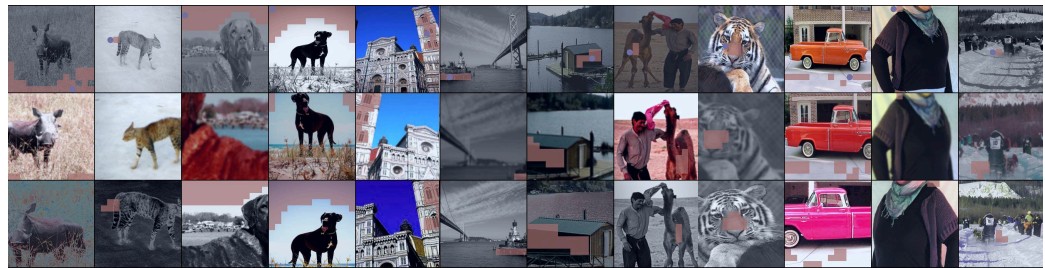

Figure 12: Discovered regions at the beginning of pretraining. The reference image (top row) shows the randomly selected seed patch (blue dot) and its associated discovered region (red). The second and third rows display the corresponding regions projected onto augmented views..

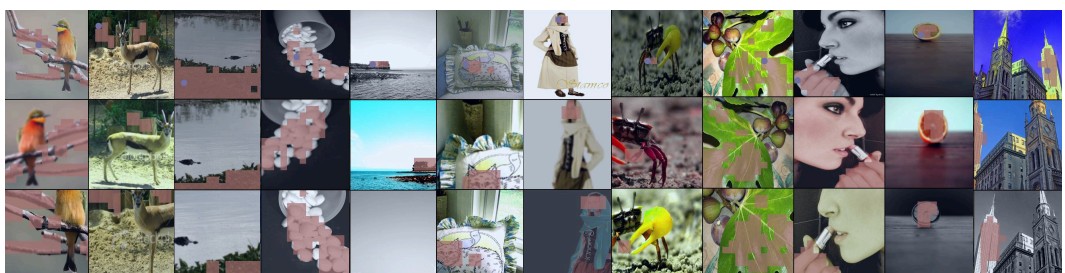

Figure 13: Discovered regions at the end of pretraining. The reference image (top row) shows the seed patch (blue dot) and the discovered region (red). The second and third rows depict the aligned regions across augmented views. Compared to the early stage, regions here are larger, semantically coherent, and align well with object parts, demonstrating the effect of CRISP's region-consistency objective.

**Evolution of Discovered Regions During Pretraining.** To better understand the dynamics of region discovery in CRISP, we compare regions obtained at the beginning and at the end of pretraining. At initialisation, when the model is warm-started from iBOT, the discovered regions are generally very small and fragmented (refer to Figure 12). This behavior is expected since the patch-level supervision in iBOT treats each patch as an independent unit, providing little incentive to aggregate them into larger, semantically consistent regions.

As training progresses under CRISP, the region-consistency objective encourages patches with correlated semantics to group together. By the end of pretraining (refer to Figure 13, the discovered regions evolve into coherent and interpretable clusters of tokens that align with meaningful parts of the scene. These clusters effectively serve as concept-level units, capturing higher-level structure beyond individual patches. This progression illustrates how CRISP transforms low-level patch features into semantically grounded region representations, highlighting the role of region-level consistency in shaping interpretable and transferable features.

A.3 TIME AND MEMORY REQUIREMENTS OF CRISP

CRISP takes about 49 minutes per epoch to pre-train a ViT-S/16 model using 4 GPUs with an effective batch size of 256, consuming 18.5 GB of memory on an NVIDIA RTX 3090. Under the same training setup, this is roughly 7 minutes longer per epoch and 2.3 GB more GPU memory than iBOT. Although CRISP is slower by approximately 0.2 epochs per hour, the additional computational cost is a reasonable trade-off for the improved spatial consistency, interpretability, and robustness of the learned representations.

