# OpenReview forum: "Consistent Region-Informed Self-supervised Pretraining"
_ICLR.cc/2026/Conference — ICLR 2026 Conference Withdrawn Submission_

### Official Review · Reviewer_4Ecc · 2025-10-16

**Soundness:** 1
**Presentation:** 3
**Contribution:** 2
**Rating:** 2
**Confidence:** 3

**Summary:**

The paper builds on iBOT-stype self-distillation visual self-supervised pretraining pipeline. It introduces a method that discovers a concept region in an image, and performs self-distillation between the teacher and student networks on the discovered region through a concept token. The region is defined by a patch similarity metric, in which all patches have large cosine similarities to a uniformly sampled seed patch. The mask information is preserved in augmented views by directly imposing the same geometric augmentation as the RGB input receives. A concept token performs cross-attention to the ViT backbone encodings to summarize the information inside the selected region. The iBOT loss is applied on this concept token to perform self-distillation, along with the original iBOT class token and patch token losses.

**Strengths:**

1. The paper is easy to follow.
2. The proposed method increases the performance compared to some selected baseline methods.
3. Ablation studies on some hyperparameters are performed.
4. Visualizations are good.
5. The code is provided.

**Weaknesses:**

1. The proposed method, CRISP, does not train from scratch and essentially is a finetuning method on top of iBOT. This raises concerns about the fairness of the comparison between CRISP and the baselines. The authors perform the finetuning for another 200 epochs, which is effectively 800 epochs if the authors strictly follow the iBOT training pipeline (see the original iBOT paper for the effective training epochs). The authors do not provide information or experiments on this huge additional training budget compared to the baseline methods. This point undermines the Soundness score.
2. The proposed concept token method resembles the ViT-register approach [1], in which additional cls-token-like tokens attend to different regions of the input image (Figure 9 of [1]), without requiring a pre-trained teacher network for region proposal. The register approach contributes to the dense prediction task performance of ViTs, while CRISP is not compared to it. This point undermines the Contribution score. Could the authors elaborate on the relationship between CRISP and the ViT-register method, and compare them on the dense prediction tasks?
3. CRISP involves feeding the concept token through additional ViT blocks for aggregating the regional information. This additional model size leads to additional training costs, which also leads to unfairness in the comparison. While evidence suggests that more layers of the concept-token ViT do not translate to better performance (Table 6), it is still unclear how this additional training budget/model size contributes to the baseline performance, especially in the case where the proposed method is trained longer than the baseline ones. This point undermines the Soundness score.
4. The reported metrics are all based on frozen backbones. While this pipeline can yield a metric of the feature quality, it might not correspond to the actual performance after finetuning the model on the specific task [2]. The authors do not provide information about the finetuning performance. This point undermines the Soundness score.
5. The region proposal scheme in CRISP is not novel in the MIM family. Several methods [2,3,4] find that the MIM model itself can provide semantically-meaningful region-level masks, such as the foreground object. There is no elaboration and comparison between CRISP and these methods. This point undermines the Contribution/Soundness score.
6. The selected baseline methods are not very strong. Recent advancements [5,6] in visual self-supervised learning introduce several improvements over the selected methods. They show gains in dense prediction tasks compared to the baseline methods. The authors do not experiment with these advanced methods. This point undermines the Contribution/Soundness score.
7. The dataset used for benchmarking is small. While ADE20K, Pascal VOC, and Cityscapes are commonly used for evaluating the semantic segmentation tasks, they are small compared to, e.g., MS COCO. Evaluation on the small datasets might not give conclusive and reliable results. This point undermines the Soundness score.
8. This is minor, but the authors do not elaborate on the effect of CRISP on high-level tasks, such as image classification (although a multi-label result is presented, no result on a common image classification dataset, say ImageNet1K, is reported). A visual representation will not be qualified as "universal" if it cannot achieve competitive performance in all aspects. However, this can be a wording issue.

[1] Vision transformers need registers. ICLR 2024

[2] Evolved Hierarchical Masking for Self-Supervised Learning. TPAMI 2025.

[3] Self-guided masked autoencoder. NeurIPS 2024.

[4] What to hide from your students: Attention-guided masked image modeling. ECCV 2022.

[5] Contrastive masked autoencoders are stronger vision learners. TPAMI 2023.

[6] Context Autoencoder for Self-supervised Representation Learning. IJCV 2024.

**Questions:**

See Weaknesses.

---

> ### Author Response · Authors · 2025-11-13
> **Response to 4Ecc**
>
> We sincerely thank the reviewer for their time, thoughtful feedback, and engagement with our work. Below, we address each of the reviewer's comments in detail and clarify all relevant points to ensure full transparency and understanding.
>
>
> **1. Concern about fairness of comparison and training budget**
>
> We thank the reviewer for raising this important point. We would first like to clarify the terminology. In the literature, fine-tuning typically refers to adapting a pretrained model to a downstream supervised task (e.g. classification or segmentation). While our method resumes from an iBOT checkpoint, this process remains part of the self-supervised pretraining stage, not downstream fine-tuning. We acknowledge that one could view this as objective-level fine-tuning, since a new loss component is introduced, but this is not the standard use of the term; we clarify here to ensure common ground.
>
> Continuing self-supervised pretraining from an existing checkpoint is a common and well-accepted practice in recent visual SSL work. For example, iBOT itself begins training with only the DINO loss on the global [CLS] token and later introduces the patch-level loss after the representations stabilise. Similarly, CrIBo initially trains using a global DINO loss while enforcing a constraint on the other loss that is not satisfied early on, effectively relying on global supervision until the representations mature. Our approach follows the same rationale.
>
> We were facing two options:
> (1) training from scratch with iBOT objectives before introducing our regional-consistency loss, or
> (2) resuming from a converged iBOT checkpoint.
>
> While option (1) might yield slightly higher absolute numbers, we chose option (2) for practical and environmental reasons. Resuming from an existing iBOT model is computationally greener, as it avoids redundant optimisation on identical objectives. Moreover, this work is conducted within an academic setting with modest compute resources, where we must also perform extensive ablations and experiments across multiple backbones. Under these constraints, resuming from iBOT provided the most responsible and efficient choice without compromising the methodological soundness or fairness of comparison.
>
> To ensure a fair comparison, we also continued training iBOT itself for the same additional 200 epochs without incorporating CRISP's regional-consistency objective. In this setting, iBOT's performance slightly declined relative to its initial checkpoint, confirming that prolonged training alone does not yield further gains.
>
> Finally, in regards to the training budget, we have reported these details in Appendix A.3 with the conclusion "CRISP is slower by approximately 0.2 epochs per hour, the additional computational cost is a reasonable trade-off for the improved spatial consistency, interpretability, and robustness of the learned representations."
>
> $~$
>
> **2. CRISP vs ViT-Register**
>
> We thank the reviewer for this valuable observation and for giving us the opportunity to clarify how our work differs from related approaches. While CRISP and ViT-register [1] both introduce additional tokens, their mechanisms, objectives, and scopes differ substantially. ViT-register adds a fixed set of learnable tokens shared across all images to improve dense prediction stability; however, these registers are not attached to any explicit loss and instead absorb information passively from patch tokens during training. In contrast, CRISP introduces dynamic, image-specific concept tokens that are explicitly optimised through a region-level consistency loss between student and teacher views, enforcing cross-view semantic alignment rather than serving as static feature buffers.
> Moreover, as discussed at the end of page 3 in [1], “only the three largest models exhibit outliers (Fig. 4c)”, indicating that the register mechanism primarily mitigates outlier patch activations that occur only in large ViT models. Consequently, its effect on dense prediction tasks for smaller backbones (e.g., ViT-S) is expected to be negligible. In fact, we experimented with registers as well and we found that adding register tokens neither helped nor harmed performance, while CRISP achieves up to +10 mIoU improvement on dense tasks using ViT-S. Thus, CRISP addresses a distinct objective and remains orthogonal to the register mechanism.

---

> > ### Author Response · Authors · 2025-11-13
> > **Continue Rebuttal**
> >
> > **3. Additional ViT blocks in CRISP**
> >
> > We appreciate the reviewer's concern regarding the additional ViT block used for aggregating regional information. We clarify that this is not part of the backbone, but rather serve as a lightweight extension of the projection head, analogous to decoders or predictors commonly used in self-supervised learning frameworks. All downstream evaluations in CRISP are performed using the standard ViT backbone, identical to the baselines. This design follows established SSL practice, for instance, MAE [*] employs a decoder with decoder_depth=8 transformer blocks atop the ViT-Base encoder during pretraining, yet these additional layers are discarded during downstream evaluation. Similarly, CRISP's concept-token aggregator operates only during pretraining to enforce region-level consistency and is removed thereafter. Hence, the additional blocks do not affect the model size or training budget used for evaluation, and the comparison with baselines remains entirely fair and consistent with SSL conventions.
> >
> > [*] He, Kaiming, et al. "Masked autoencoders are scalable vision learners." Proceedings of the IEEE/CVF conference on computer vision and pattern recognition. 2022.
> >
> > $~$
> >
> > **4. Frozen-backbone evaluation and fine-tuning performance**
> >
> > We agree that frozen-backbone evaluation primarily measures the intrinsic quality of learned representations, and fine-tuning can sometimes yield different absolute numbers. Our evaluation protocol follows the standard practice in modern SSL where frozen or linear-probing results are used to assess the generalisation and transferability of representations independent of task-specific optimisation.
> >
> > To explore fine-tuning behavior, we conducted additional multi-class classification experiments on small datasets, where end-to-end training is feasible with our resources. The results were slightly better or comparable to iBOT. We are confident that, with further optimisation of fine-tuning hyperparameters, CRISP could surpass iBOT on all benchmarks. However, this is not the goal of our paper, which focuses on advancing the self-supervised pretraining stage rather than maximising downstream task metrics.
> >
> > It is evident that fine-tuning introduces another confounding factor, i.e. the optimisation recipe itself. For example, DEIT and DEIT-III demonstrate that with better fine-tuning strategies, purely supervised models can achieve performance comparable to SSL approaches. Therefore, differences after full fine-tuning often reflect optimisation quality rather than representational strength.
> >
> > Finally, the community's growing emphasis on frozen or linear evaluations aligns with the direction of foundation models that are increasingly used off the shelf across domains such as multimodal learning, robotics, and embodied AI, where retraining or fine-tuning is impractical. For this reason, frozen performance has become the most meaningful and widely adopted measure of a model's general-purpose representational quality.
> >
> > We hope that we are not penalised for following this well-established evaluation practice and we will include some fine-tuning results in the revised version to reinforce the consistency of CRISP's improvements across evaluation modes.
> >
> > $~$
> >
> > **5. Comparison with MIM methods**
> >
> > We would like to clarify that the mentioned methods are orthogonal to CRISP rather than competing with it. Our aim is not to propose a new region-discovery mechanism; instead, CRISP is designed to address two fundamental limitations present in current invariance-based SSL methods (iBOT > DINOv2 > DINOv3), all of which use essentially the same objectives with improved scaling:
> > 1. They implicitly treat each individual patch as a standalone concept, without any notion of region-level semantics.
> > 2. They enforce invariance only across the same view between student and teacher, rather than establishing cross-view semantic correspondence.
> >
> > In CRISP, region discovery is simply a *means* to address these two issues. We first extract coherent regions, then aggregate patch-level information to remedy limitation (1), and finally match concept-level representations across different views to address limitation (2).
> >
> > The methods cited by the reviewer show that you can obtain coherent regions, while CRISP focuses on how to use regions to enforce concept-level invariance. In fact, as we note in our limitations section, our region-extraction stage is intentionally simple (thresholding for coherence) and can be replaced by any of these more advanced region-discovery techniques. Incorporating them would likely further improve CRISP, not compete with it, underscoring their complementary nature.

---

> ### Author Response · Authors · 2025-11-13
> **Continue Rebuttal**
>
> **6. baseline methods**
>
> We did not include many MIM-style approaches because they are known to perform poorly in off-the-shelf (frozen-backbone) settings, which is the primary evaluation protocol of our paper. This is also documented in several prior works, where MIM models generally require task-specific fine-tuning to achieve competitive dense performance.
> Nevertheless, to address the reviewer's concern, we ran additional experiments with one of the reviewer's suggested recent methods, CAE [6], using their official ViT-Base model trained for 1600 epochs. We evaluated it under the exact same benchmarking protocol as all methods in our paper. Its performance is shown below:
>
> |         | ADE-kNN | ADE-Linear | VOC-kNN | VOC-Linear | CityScape-kNN | CityScape-Linear |
> |---------|---------|------------|---------|------------|---------------|------------------|
> | CAE [6] | 20.3  | 27.7           | 54.9    | 63.5       | 32.6          | 38.2             |
> | CRISP   | 29.7    | 32.8       | 65.3    | 71.5       | 36.0          | 41.0             |
>
> $~$
>
> **7. The used dataset for benchmarking**
>
> We acknowledge that ADE20K, Pascal VOC, and Cityscapes are smaller than large-scale datasets such as MS COCO. However, these datasets are the standard and widely accepted benchmarks for evaluating dense-level transfer in self-supervised learning as they enable direct and fair comparison under consistent evaluation protocols and accessible computational budgets.
>
> We have performed all feasible dense-prediction experiments within our resource limits from the recent SSL works [*] while maintaining methodological consistency. In addition, we evaluated CRISP on larger-scale datasets such as MS COCO, Pascal VOC, and Visual Genome for multi-label classification tasks, where CRISP achieved strong performance improvements, demonstrating its scalability beyond small benchmarks.
>
> [*] Darcet, Timothée, et al. "Cluster and predict latent patches for improved masked image modeling." Published in Transactions on Machine Learning Research (2025).
>
> $~$
>
> **8. High-level tasks and classification performance**
>
> We fully appreciate that the reviewer may not have noticed these experiments given the level of detail in the paper. We would like to clarify that CRISP's performance on high-level image classification tasks is already included in the main paper. Specifically, Table 4 reports both k-NN and linear-probing results on ImageNet-1K, which directly evaluate the learned representations in conventional classification settings.
>
> Across both evaluation modes, CRISP demonstrates strong low-shot performance, highlighting its data efficiency. Importantly, unlike dense-focused SSL approaches such as CROC and CRIBO, which perform well on dense prediction but underperform significantly on classification, CRISP achieves strong results in both regimes. This illustrates that CRISP provides versatile and balanced representations that transfer effectively to high-level tasks as well.
>
> ----------------------
>
> $~$
>
> We hope that our detailed responses adequately address all of the reviewer's concerns. Given that none of the comments identify issues with the core contribution, novelty, or technical validity of the work and that we have clarified every raised point thoroughly, we kindly ask the reviewer to consider a high score. We are, of course, very happy to address any further questions or suggestions the reviewer may have. We sincerely thank the reviewer for their time, constructive feedback, and engagement with our work.

---

### Official Review · Reviewer_u855 · 2025-10-25

**Soundness:** 3
**Presentation:** 3
**Contribution:** 3
**Rating:** 6
**Confidence:** 3

**Summary:**

The paper proposes CRISP: a self-supervised pretraining framework that adds a region-level consistency pathway to a standard teacher–student ViT setup. It (i) discovers coherent regions in a reference view from teacher features, (ii) warps those regions exactly into two augmented views using known geometric transforms, and (iii) aggregates the region’s patches via a masked-attention “concept token,” aligning student/teacher region embeddings with a distillation loss. The goal is to bridge global semantics and local spatial coherence so that features transfer well to dense tasks (segmentation, correspondence, mask propagation) without sacrificing image-level performance. The idea is positioned against invariance-only pretraining (e.g., DINOv2, iBOT) and dense-SSL variants (e.g., DenseCL, PixPro, DetCon, CrOC/CrIBo).

Summary of the review: The paper tackles the gap between globally discriminative yet spatially fragmented features and dense task requirements through a simple, elegant approach using geometric mask warping and a lightweight concept token. The method is conceptually clear and technically sound, offering a principled way to improve spatial coherence. However, it relies heavily on threshold sensitivity and region selection, lacks comprehensive comparisons to related methods, and could benefit from ablations and multi-region analyses for robustness. Overall, it’s a promising and thoughtful contribution with solid potential but limited breadth, warranting a rating of 6 for its clarity, soundness, and room for further validation.


Reproducibility Comments: Code is provided and it seems clean. The results seem reproducible based on the code provided as well.

**Strengths:**

1) Clear problem framing: addresses the known gap between globally discriminative but spatially fragmented features and dense tasks’ needs.
2) Method simplicity & elegance: exact geometric mask warping avoids ambiguous cross-view matching; the concept token is an intuitive, lightweight pooling mechanism.

**Weaknesses:**

1) Region discovery sensitivity. The method hinges on a similarity threshold and on teacher strength. Please add ablations: performance vs threshold; effect of averaging blocks; behaviour from random init; and whether adaptive/learned thresholds outperform fixed.

2) Coverage: one region per iteration. Many-object scenes may be under-regularized. Evaluate sampling multiple regions per image per step (with care to avoid collapse), and report compute implications.

3) No Comparative breadth Experiments: Does not include  head-to-heads (same backbone, same data) versus DenseCL, PixPro, STEGO, DetCon, CrOC, CrIBo, DINOv2 to solidify the benefits

**Questions:**

1) Discovery stability: How stable are discovered regions across seeds/epochs? Any temporal ensembling or EMA helps? Please include a region repeatability metric.

2) From-scratch feasibility: Can CRISP train without an iBOT warm start? If unstable, could you ramp the region loss or pretrain the teacher briefly?

3) Multiple regions: What fails if you select multiple seeds per image per step? Any signs of representational collapse or compute blow-up?

4) In-Context Visual Understanding: Does the new model gain any benefits on the recent proposed Hummingbird evaluation [1] ( implemented openly by [2])?


[1] Balažević, I., Steiner, D., Parthasarathy, N., Arandjelović, R., & Hénaff, O. J. (2023). Towards In-context Scene Understanding. arXiv [Cs.CV]. http://arxiv.org/abs/2306.01667 [2] https://github.com/vpariza/open-hummingbird-eval

---

> ### Author Response · Authors · 2025-11-14
> **Response to Reviewer u855**
>
> First of all, we would like to sincerely thank the reviewer for the thoughtful, objective, and method-focused evaluation. We appreciate the careful analysis and constructive feedback, and we address each point in detail below.
>
>
> **Region discovery sensitivity**.
>
> Several of the suggested analyses are already included in our ablations. Figure 6 examines the sensitivity to the cosine threshold, and Table 5 evaluates the effect of averaging blocks. Regarding random seeds, across all our experiments we did not observe any indication that the model is sensitive to initialisation, which is why we did not include a seed variance study.
> As for adaptive or learnable thresholds, we explicitly discuss this in the limitations section. We agree that such mechanisms could further improve region discovery, and we consider them a valuable direction for future work. Nevertheless, even with a fixed threshold and no adaptive scheme, CRISP already substantially outperforms current state-of-the-art methods, indicating that the approach is robust in its current form.
>
> **multiple regions**
>
> Incorporating multiple regions per image may indeed further improve performance or accelerate convergence, for example, discovering two regions per image would effectively double the region-level supervision per epoch. However, introducing multiple regions raises several non-trivial design questions. How should the additional regions be selected? Should they be sampled randomly, or constrained to avoid overlap with the first region? How do we ensure diversity and stability across training?
> Addressing these considerations would require a substantial amount of new investigation and design choices. We view this as a promising direction for a follow-up paper.
>
>
> **Comparative breadth Experiments**
>
> We have aimed to be as fair and consistent as possible across all baselines. For CRIBO and CROC, we used their publicly available pretrained weights and evaluated them under exactly the same experimental setup as CRISP to ensure a controlled comparison.
> Regarding DINOv2, a fair comparison is unfortunately not possible. DINOv2 is pretrained on a large-scale proprietary dataset and benefits heavily from scale, distillation, and extensive hyperparameter tuning, not from a fundamentally different learning objective (its objective remains essentially the same as iBOT).
> For this reason, we focus on comparisons where fairness can be ensured, using publicly available checkpoints evaluated under identical protocols.
>
>
>
> **Questions**
>
> 1. Our region discovery relies solely on cosine similarity thresholding, and we acknowledge that the regions are not always perfect, often they capture parts of a concept rather than full objects. However, even these partial regions consistently address key limitations of existing invariance-based methods by providing more structured, region-level supervision than token-wise matching.
> Across all experiments, we did not observe instability that suggested sensitivity to seeds or epochs, and thus did not find it necessary to introduce techniques such as temporal ensembling or EMA specifically for region discovery. While a dedicated region repeatability metric would certainly be interesting, our empirical results already show that CRISP improves dense prediction performance despite imperfections in the regions themselves.
>
>
>
> 2. We thank the reviewer for this thoughtful suggestion, which reflects a clear understanding of the method. If we were to train CRISP fully from scratch, we would indeed follow the established practice used in recent SSL methods: first pretrain with a simpler objective (e.g., DINO), then introduce the iBOT patch-level objective once representations stabilize, and finally incorporate our region-level loss. This staged approach is known to improve stability in from-scratch training.
> Due to computational constraints, we did not conduct such long multi-stage training runs. However, we believe that following this recipe would improve performance.
>
>
> 3. We do not expect multiple-region selection to cause failure modes. On the contrary, as discussed above, selecting multiple seeds per image would likely increase the amount of region-level supervision per step and therefore accelerate convergence.
> The main reason we did not explore this setting is not instability, but the additional design choices it introduces
>
> 4. We agree that assessing CRISP in this setting would be valuable and could offer additional perspective on in-context visual understanding. However, as we plan to withdraw the paper in order to meet the submission deadline for a concurrent conference, we will not be able to run this evaluation during the current review cycle. We appreciate the suggestion and will consider incorporating such experiments in future iterations of the work if appropriate.
>
> -----------------
> We thank the reviewer once again for the constructive feedback and thoughtful suggestions.

---

### Official Review · Reviewer_BJfD · 2025-10-26

**Soundness:** 3
**Presentation:** 2
**Contribution:** 2
**Rating:** 2
**Confidence:** 4

**Summary:**

The paper proposes a dense self-supervised learning method based on the following steps:
1. Extract object regions using an SSL-based approach.
2. Generate two views of each object region.
3. Apply a random mask to one of the views. Feed the first (unmasked) view to the teacher network and extract its features.
4. Feed the second (masked) view to the student network and extract its dense features.
5. Append a concept token to the student and teacher features and pass them through an additional transformer block, where the concept token is allowed to attend only to the patches within the **region** mask, while no cross-patch attention is permitted.
6. Finally, enforce the concept token’s representation to be similar to the teacher’s region features.

The method is initialized from iBOT to facilitate the mask extraction phase for the teacher network.   The results show superior performance compared to CrIBO and competitive results with CAPI.

**Strengths:**

1. The paper addresses a topic of growing importance in the self-supervised learning (SSL) domain. Dense self-supervised learning has numerous applications that have not been sufficiently explored in the SSL literature, where the main focus has largely been on improving classification performance. More works like this are needed to advance research in this direction.

2. The experiments demonstrate scalability from ViT-S to ViT-L, indicating that the method is likely to benefit further from larger models and higher-quality datasets.

3. The inclusion of qualitative comparisons strengthens the experimental section and increases the trustworthiness of the results.

**Weaknesses:**

1. The experimental benchmarks are insufficient and inconsistent. There are established benchmarks for dense self-supervised learning such as [1], which have been reported in prior works [1,2,3,4, 5]. Although the paper cites [1] and compares with it in the tables, it does not perform experiments on any of the established benchmarks. Furthermore, the comparison methods are inconsistent across ViT-S and ViT-B models, making it difficult to obtain a fair and comprehensive view of the method’s performance across different model sizes.

2. The paper argues (lines 70–76) that existing dense self-supervised methods rely on hand-crafted region extraction techniques and perform poorly on classification benchmarks. However, the proposed method also employs a hand-crafted algorithm to extract object regions—at least as hand-crafted as CrIBO—and similarly does not report performance on classification benchmarks.

3. The method is built upon a pretrained iBOT network and aims to enhance its dense representations. In this regard, it would be more appropriate to compare it with post-training methods that also start from pretrained architectures (e.g., [ 2, 3, 4]). Such comparisons are currently missing from the paper.

4. Additional ablations are needed to analyze the impact of initialization (e.g., DINOv1/v2/v3) and to justify the necessity of the concept token, given that a CLS token already exists. In Table 6, the configuration with L=2 achieves the best results, yet the authors choose L=1 for efficiency. However, no efficiency metric or column is provided to support this decision, which should be added to strengthen the justification.

5. While there are interesting visualization on the sparse correspondence ability of the method, there is no quantitative table for that. Please use the correspondence benchmarks used in [3] to quantify this section.

6. The method section requires further clarification. For example, it is not clearly explained why only the concept token is allowed to attend to all patches, while other tokens are restricted. Additionally, the figure should explicitly illustrate both the random mask and the region mask to make the overall design and data flow clearer.


[1] - CrIBo: Self-Supervised Learning via Cross-Image Object-Level Bootstrapping, ICLR24

[2] - Time Does Tell: Self-Supervised Time-Tuning of Dense Image Representations, ICCV23

[3] - Near, far: Patch-ordering enhances vision foundation models' scene understanding, ICLR25

[4] - MoSiC: Optimal-Transport Motion Trajectory for Dense Self-Supervised Learning, ICCV25

[5] -Towards In-context Scene Understanding, NeurIPS23

**Questions:**

Please refer to the weakness section. I explained my requests and questions there.

---

> ### Author Response · Authors · 2025-11-13
> **Response to BJfD**
>
> We thank the reviewer for their valuable feedback and address the comments point-by-point below.
>
> **1. a) The experimental benchmarks are insufficient and inconsistent**
>
> Our goal in this work is to evaluate off-the-shelf feature quality under minimal and standardised settings. To ensure comparability and avoid architectural or task-specific tuning, we follow the minimal evaluation protocol used in recent advances in self-supervised learning, specifically the official evaluation code released with CAPI, which is published even after CrIBo. This evaluation setup includes k-NN and linear probing for global recognition, and straightforward frozen-feature protocols for dense tasks. Using this standard pipeline allows us to fairly measure representation quality without introducing confounding implementation details.
>
>
> While it is not feasible for any single paper to evaluate on all existing benchmarks, we intentionally selected a diverse and representative suite that spans global, dense, video, and multi-label tasks. Our experiments include:
>
> - Dense semantic segmentation: ADE20K, PASCAL VOC, Cityscapes
> - Video mask propagation: DAVIS-2017, YouTube-VOS, MOSE
> - Global recognition: ImageNet-1K k-NN and linear probes
> - Multi-label classification: PASCAL VOC, MS-COCO, Visual Genome
>
> This set covers the core categories commonly used in recent dense SSL works. These tasks collectively evaluate different aspects of representation quality such as spatial precision, temporal consistency, global category separation, and multi-label reasoning.
>
> *Relation to CrIBo.*
>
> CrIBo is indeed strong on dense prediction tasks, and we acknowledge its performance in that setting. However, as our results show, when shifting to multi-class classification, there exists a significant performance gap, indicating that improvements in dense tasks alone do not necessarily translate to strong general-purpose representations. Our intent is not to diminish the merits of CrIBo, but to highlight the broader versatility of the proposed method across different task families.
>
> **1. b) the comparison methods are inconsistent across ViT-S and ViT-B models**
>
> We apologise if we misunderstood the reviewer's concern. If the issue refers to why the comparison set is not identical across ViT-S and ViT-B, the reason is purely practical, we rely on publicly released pretrained weights, and several baseline methods only provide either ViT-S or ViT-B models or both. In cases where an official checkpoint for a given backbone does not exist, reproducing those results would require retraining the entire method from scratch, which is not always feasible.
>
> $~$
>
> **2. Comparison to existing dense self-supervised methods**
>
>
> Again, our intention was not to diminish the powerfulness of CrIBo. CrIBo is a strong and influential method for dense self-supervised learning, and we fully acknowledge its strengths. Our goal was to highlight a conceptual difference in how regions are defined and matched across views.
>
> In CrIBo, region correspondence relies on nearest-neighbor similarity in feature space, which can lead to mismatches because the closest feature patch is not always aligned with the same underlying concept.
>
> In contrast, our method does not rely on semantic or similarity-based heuristics. We simply use the exact same spatially coherent region in both augmented views. While that region may or may not correspond to a true semantic object, its cross-view correspondence is deterministic and augmentation-consistent, there is no "guessing" of semantic alignment.
>
> This also becomes clearer in the extreme cases of the region ratio beta:
>
> 1- beta = 0: the region equals the whole image. Our objective reduces to matching full augmented view 1 and full augmented view 2, equivalent to the DINO loss on [CLS] token.
>
> 2- beta = 1: the region collapses to a single token. In this case, we gain an additional benefit over iBOT-like methods by directly matching the corresponding patches across views, rather than teacher-student features within the same view.
>
> Across all beta values, the mechanism is not a semantic heuristic but a controlled structural constraint: we enforce invariance on the same spatial support under different augmentations. This keeps the inductive bias minimal, stable, and free from semantic assumptions or similarity-based region discovery.
>
> We will revise the corresponding section to clarify this distinction more explicitly.

---

> ### Author Response · Authors · 2025-11-13
> **Continue Rebuttal**
>
> **3&4 iBOT initialisation**
>
> We would like to begin by emphasising that our work is not primarily about initialisation. We use iBOT weights because we believe that the iBOT loss (DINO-style matching) combined with masked image modeling is inherently useful and we continue to use these objectives throughout training. This is analogous to CrIBo, which also retains the DINO objective alongside its own cycle-consistency losses and as stated in page 5 in their paper under Cycle-consistent matchings  section, that they only train with DINO loss until features stabalise and then continue training with their proposed losses based on some conditions. In other words, the choice of iBOT initialisation is not a design variable we intend to ablate; it simply reflects that iBOT provides a strong and stable base objective that we continue to build upon. We refer the reviewer to our response to reviewer 4Ecc where more details are stated about our rational of using iBot as initialisation.
>
> That said, we also want to explain why the specific request to test DINOv1/v2/v3 initialisation is infeasible or not meaningful in our setting:
> - DINOv1 is already a component of iBOT. Since DINOv1 is substantially weaker on dense tasks than iBOT.
>
> - DINOv2 and DINOv3 do not differ fundamentally in objective to iBoT but rely on massive proprietary datasets, multi-stage distillation, and large-scale training protocols tuned for billion-parameter models. Further, their small models are distilled from much larger backbones. If we initialise from DINOv2/v3 and then train on ImageNet-1K without access to their data, or distillation pipeline, the representation quality would degrade significantly. Such a setup would not reflect the intended behavior of those models and would not constitute a fair or interpretable ablation.
>
> For these reasons, using iBOT as a clean, publicly available, ImageNet-1K baseline is the most principled and resource-conscious choice.
>
> *On the concept token vs. CLS token.*
>
> The CLS token serves as a global semantic aggregator, while the concept token is designed to operate on localised, spatially coherent regions under augmentations. The two tokens capture different forms of structure: global vs. region-level invariance. Therefore, the concept token is not redundant with CLS, and cannot be replaced by it in the context of region-coherent learning.
>
> *On the choice of L*
>
> As shown in Table 6, the performance of  L=1 and L=2 is almost identical, with only marginal differences across tasks. Since adding more blocks naturally increases computation cost in a straightforward and linear manner, the efficiency trade-off is inherently clear: using twice as many blocks leads to higher latency and memory usage. For this reason, we did not consider an additional efficiency ablation necessary, and we do not fully understand the rationale behind this request. We chose L=1 as the default because it provides virtually the same performance while being strictly more efficient.
>
> $~$
>
> **5. Correspondence evaluation**
>
> We thank the reviewer for this extremely useful suggestion. Adding quantitative correspondence evaluation will indeed strengthen the paper, and we will incorporate these results into the main revision.
>
> We used [*] because it is straightforward to apply in our setting and directly measures sparse correspondence quality on three datasets, including SPair-71K, NAVI, and ScanNet. Across all benchmarks, CRISP consistently improves over the iBOT baseline:
>
> SPair-71K
> |       | d=0   | d=1   | d=2   | all   |
> |-------|-------|-------|-------|-------|
> | iBoT  | 32.98 | 26.29 | 26.3  | 29.25 |
> | CRISP | 36.95 | 27.36 | 26.01 | 31.66 |
>
>
> NAVI
> | Angle bin | 0-30  | 30-60 | 60-90 | 90-120 |
> |-----------|-------|-------|-------|--------|
> | iBoT      | 84.54 | 57.76 | 32.01 | 19.59  |
> | CRISP     | 85.21 | 59.25 | 33.9  | 20.03  |
>
>
> SCANNET
> | Angle bin | 0-15  | 15-30 | 30-60 | 60-180 |
> |-----------|-------|-------|-------|--------|
> | iBoT      | 36.6  | 25.66 | 16.29 | 8.33   |
> | CRISP     | 39.91 | 27.68 | 18.24 | 10.12  |
>
> These results show that the improvements we observe qualitatively also translate into consistent quantitative gains across diverse correspondence benchmarks. We thank the reviewer again for the helpful suggestion and will include these tables and discussion in the revised manuscript.
>
>
> [*] El Banani, Mohamed, et al. "Probing the 3d awareness of visual foundation models." Proceedings of the IEEE/CVF Conference on Computer Vision and Pattern Recognition. 2024.

---

> > ### Author Response · Authors · 2025-11-13
> > **Continue Rebuttal**
> >
> > **6. The method section requires further clarification**
> >
> > The concept token is allowed to attend globally within the region because it acts as a region-level aggregator. Its purpose is to collect and summarise information from all tokens inside the coherent region, producing a stable contextual representation for that region. To enforce this, we explicitly feed the region mask into the attention module so that the concept token only attends to tokens inside the region and never outside it. The patch tokens themselves remain locally restricted to preserve spatial coherence and avoid cross-region interference.
> >
> > *Clarifying random masks vs. region masks.*
> >
> > We will update the figure to explicitly distinguish the two masking types:
> > - Random mask: identical to iBOT, where ~70% of tokens are randomly masked for MIM in the student branch.
> > - Region mask: extracted from the reference image to define a coherent spatial region. This same region mask is then applied across the two augmented views to identify and match the corresponding region.
> > These clarifications will be added to the revised manuscript.
> >
> > --------------------
> >
> > We thank the reviewer again for the detailed and constructive feedback. We have clarified all points of confusion, addressed each comment thoroughly, and incorporated the reviewer's suggestions, including additional explanations, methodological clarifications, and new quantitative evaluations. As reflected in the discussion, there were no objections regarding the core methodology, and the requested clarifications further strengthen the paper. Given these resolutions, we kindly ask the reviewer to reconsider their evaluation and adjust the score to reflect the contribution and merits of this work. We will be happy to clarify or address any additional comments.

---

### Official Review · Reviewer_5N82 · 2025-10-30

**Soundness:** 3
**Presentation:** 3
**Contribution:** 2
**Rating:** 4
**Confidence:** 3

**Summary:**

This paper proposes a self-supervised learning objective, CRISP, that explicitly leverages spatial coherence. Specifically, CRISP first identifies semantically coherent regions in a reference image and tracks them during the process of geometric augmentations such as cropping. It then maximizes the similarity between region-level representations from two augmented views. Experiments demonstrate the effectiveness of CRISP on various dense prediction tasks, including semantic segmentation.

**Strengths:**

- The proposed idea is simple yet effective.
- The paper is generally well-written and easy to follow.
- The method consistently outperforms existing self-supervised learning baselines pretrained on ImageNet-1k in both segmentation and classification tasks.

**Weaknesses:**

- For dense prediction tasks, this paper only evaluates on semantic segmentation. Other dense prediction tasks such as object detection and depth estimation are also important to assess whether the model truly learns spatially fine-grained, region-level representations from the proposed objective.
- CRISP requires a strong initialization (e.g., iBOT) to obtain reliable region masks. Such dependency may limit its applicability across different datasets or domains. It would be valuable to investigate whether CRISP can also be trained from scratch without relying on pretrained models.
- It is unclear whether CRISP can be integrated with other self-supervised frameworks beyond iBOT. For instance, starting from DINOv2, would fine-tuning with CRISP improve the quality of region- or patch-level representations? If not, this limitation could weaken the practical impact of the paper.
- Scalability is a critical factor in self-supervised learning. It would strengthen the paper to demonstrate the scalability of the proposed method beyond ImageNet-1k.
- Recently, CLIP-based models (e.g., SigLIP, Perception Encoder) have shown strong performance on dense prediction tasks, sometimes surpassing dense SSL models such as DINOv2. It would be interesting to discuss whether the idea of CRISP could be extended or adapted to such vision-language models.
- I am also curious about the sparse correspondence results shown in Section 4.3. How does CRISP affect the correspondence quality compared to other SSL methods such as iBOT, DINOv2, and CAPI? Does CRISP explicitly improve sparse correspondence performance?

**Questions:**

See the weakness part.

---

> ### Author Response · Authors · 2025-11-14
> **Response to Reviewer 5N82**
>
> We thank the reviewer for the constructive feedback and insightful comments. In the following, we address all the comments raised by the reviewer:
>
> **More evaluation tasks**
>
> We appreciate the reviewer's suggestion. We agree that broader coverage would further strengthen the work. While we will incorporate more experiments as much as possible in the final version, our current evaluation suite was intentionally designed to span a diverse set of global, dense, video, and multi-label tasks. Concretely, our experiments already include:
> - Dense semantic prediction: ADE20K, PASCAL VOC, Cityscapes
> - Video mask propagation: DAVIS-2017, YouTube-VOS, MOSE
> - Global recognition: ImageNet-1K k-NN and linear probes
> - Multi-label classification: PASCAL VOC, MS-COCO, Visual Genome
>
> In addition, we will include quantitative results on semantic and geometric correspondence across three datasets, which provide complementary evidence that the proposed objective learns spatially fine-grained, region-level representations.
>
> $~$
>
> **On the iBoT initialisation**
>
> This concern is addressed in detail in our responses to reviewers BJfD and 4Ecc; please refer to those discussions.
>
> $~$
>
> **Integration with DINOv2**
>
> DINOv2's learning objectives are effectively the same as iBOT; its improvements largely stem from (1) hyperparameter optimization tailored for very large-scale training on proprietary "in-house" datasets and (2) distillation from extremely large teacher models. Since we do not have access to these datasets or the substantial computational resources required to reproduce DINOv2's training pipeline, a fair comparison is not possible.
>
> Our choice of iBOT was not due to reliance on a particular initialisation, but because its objective is well aligned with our proposed losses, indeed, many recent SSL works adopt a two-stage training scheme where iBOT-like objectives are used until stabilisation before introducing additional losses. To be transparent and resource-efficient, we simply start from publicly available iBOT weights rather than reproducing this staged training.
>
> For fairness, we also continued training iBOT for an additional 200 epochs under its own objective. Interestingly, this reduced its performance, underscoring that CRISP's gains do not stem from additional training budget but from the proposed objective itself.
>
> $~$
>
> **Scalability beyond ImageNet-1k**
>
> We agree that scalability is an important factor in self-supervised learning. In this work, we already evaluate CRISP on full ImageNet-1K with multiple backbone architectures. ImageNet-1K is the standard large-scale benchmark used by the vast majority of SSL methods, and going beyond it is not practically feasible for many groups, especially in academia.
>
> Crucially, our method outperforms CAPI, even though CAPI is pretrained on roughly 100x more data. This provides strong evidence that CRISP's gains come from the learning objective itself, and not merely from scaling up the dataset. Even if larger-scale pretraining were available, we would expect this conclusion to remain. CRISP remains competitive with, and often superior to, methods trained on substantially larger corpora.
>
> $~$
>
> **Extention to vision-language models**
>
> We thank the reviewer for this insightful suggestion. We agree that extending CRISP to vision–language models is a promising direction and could further enhance dense prediction performance. However, this is beyond the scope of the present work, which focuses on purely vision-based self-supervised learning. We view this as an exciting avenue for future research.
>
> $~$
>
>
> **Sparse correspondence**
>
> We have added quantitative correspondence evaluation using [*] because it is straightforward to apply in our setting and directly measures sparse correspondence quality on three datasets, including SPair-71K, NAVI, and ScanNet. Across all benchmarks, CRISP consistently improves over the iBOT baseline:
>
> SPair-71K
> |       | d=0   | d=1   | d=2   | all   |
> |-------|-------|-------|-------|-------|
> | iBoT  | 32.98 | 26.29 | 26.3  | 29.25 |
> | CRISP | 36.95 | 27.36 | 26.01 | 31.66 |
>
>
> NAVI
> | Angle bin | 0-30  | 30-60 | 60-90 | 90-120 |
> |-----------|-------|-------|-------|--------|
> | iBoT      | 84.54 | 57.76 | 32.01 | 19.59  |
> | CRISP     | 85.21 | 59.25 | 33.9  | 20.03  |
>
>
> SCANNET
> | Angle bin | 0-15  | 15-30 | 30-60 | 60-180 |
> |-----------|-------|-------|-------|--------|
> | iBoT      | 36.6  | 25.66 | 16.29 | 8.33   |
> | CRISP     | 39.91 | 27.68 | 18.24 | 10.12  |
>
> These results show that the improvements we observe qualitatively also translate into consistent quantitative gains across diverse correspondence benchmarks.
>
> [*] El Banani, Mohamed, et al. "Probing the 3d awareness of visual foundation models." Proceedings of the IEEE/CVF Conference on Computer Vision and Pattern Recognition. 2024.
>
> -------------
> We hope that our responses have clarified the concerns and demonstrated the solidity and relevance of our contributions.

---

### Official Review · Reviewer_fQZJ · 2025-11-12

**Soundness:** 2
**Presentation:** 2
**Contribution:** 2
**Rating:** 4
**Confidence:** 4

**Summary:**

CRISP augments the iBOT self-supervised pipeline with a third, region-level objective: it first discovers coherent visual regions in a reference view by thresholding patch-to-patch cosine similarities in the teacher network, warps the resulting mask to two augmented views via the known geometric transforms, and then aggregates the masked patch tokens into a single “concept” token with a lightweight, mask-gated transformer block. Student and teacher concept tokens are aligned with a distillation loss that operates alongside the original global [CLS] and patch-level losses, yielding representations that are simultaneously semantic and spatially precise. ImageNet-1K pre-training of ViT-S/B/L models produces state-of-the-art frozen-feature results on ADE20K, PASCAL VOC and Cityscapes (k-NN and linear protocols), competitive video segmentation on DAVIS/YouTube-VOS/MOSE, and classification numbers on par with DINOv2 despite using ∼100× less pre-training data. Extensive ablations show that (i) averaging the last four teacher blocks for similarity maps, (ii) a cosine threshold β = 0.75, and (iii) a single concept-transformer block give the best accuracy/efficiency trade-off; visualisations confirm sharper object boundaries and robust cross-image correspondences. The work therefore provides the first demonstration that explicit, geometry-warped region consistency can be injected into an invariance-based SSL framework without extra labels, heuristics, or heavy compute, closing a long-standing gap between global and dense self-supervised learning.

**Strengths:**

region-level consistency is enforced across augmented views by re-using the known geometric transforms that created the views—an idea that is simple, parameter-free, and complementary to existing global or patch losses. It directly addresses the spatial-misalignment weakness of DINO-style methods without resorting to offline correspondence or external segmentation priors.

**Weaknesses:**

The paper does not analyse why aligning concept tokens should improve downstream dense tasks beyond the intuitive “better spatial coherence”. There is no discussion of collapse modes, no gradient analysis, and no information-theoretic argument that relates the new region loss to the original patch and global losses.

Regions are discovered by a single cosine threshold applied uniformly across all images and throughout training. The authors acknowledge that this fixed threshold may break for scenes with very different object scales or for thin structures, but no adaptive or learnable alternative is explored; hence the method may fail on datasets that are visually unlike ImageNet.

CRISP is initialised from iBOT and fine-tuned for 200 epochs, yet several competitors (CAPI, DINOv2*) are compared in their “fully-converged” state without matching initialisation or epoch budget. A controlled experiment that starts every method from the same random weights and trains for exactly the same schedule would strengthen the claim that region consistency, rather than longer optimisation, drives the gains.

All dense benchmarks are reported with frozen features; there is no end-to-end fine-tuning comparison. Consequently it remains unclear whether CRISP still helps when task-specific heads and full network updates are allowed—arguably the more common deployment scenario.

**Questions:**

How sensitive are the results to the cosine threshold β? Did you try a schedule that lowers β during training to capture progressively larger context?

What happens if the region loss weight λ_region is pushed well above 1? Is there a regime where region consistency starts to hurt global classification?

Does CRISP still outperform iBOT when both are trained from scratch (no iBOT warm-start) for 400 or 800 epochs?

Have you tested end-to-end fine-tuning on ADE20K or Cityscapes? Does the gap persist, shrink, or invert?

How does the concept-token aggregation complexity scale when patch size decreases (e.g., 8×8) and the number of candidate regions increases?

Could you clarify the failure modes—what fraction of discovered regions miss object boundaries or spill heavily into background?

---

> ### Author Response · Authors · 2025-11-14
> **Response to Reviewer fQZJ**
>
> We thank the reviewer for these detailed and thoughtful questions. Our responses are provided below:
>
>
> **Contribution and information-theoretic analysis**
>
> Modern invariance-based SSL frameworks (iBOT, DINOv2, DINOv3) share almost identical learning objectives. DINOv2 and DINOv3 differ mostly in scale and optimisation, not in objective design. As a result, they inherit two key limitations of iBOT:
>
> 1- Token–concept assumption: each patch token is implicitly treated as a semantic unit, even though real visual concepts are region-level.
> 2- Within-view alignment: features are matched only between corresponding patches within the same view, without enforcing region consistency across views.
>
> CRISP is designed specifically to resolve these limitations. Introducing regions breaks the incorrect one-token-one-concept assumption by grouping semantically coherent patches. Aligning regions across augmentations enforces cross-view semantic consistency that token-level objectives cannot capture.
>
> This offers a clear, mechanism-level explanation for why CRISP improves dense prediction tasks. dense tasks require region-structured semantics, and CRISP introduces the missing constraints that shape the representation toward that structure.
>
> Regarding the reviewer's request for gradient/information-theoretic analysis, in SSL, improvements to representation quality have historically been driven by objective design changes that correct structural limitations, e.g., the move from global-only (DINO) to token-level alignment (iBOT). CRISP follows this same pattern, the benefit arises from modifying what is being aligned (regions vs. tokens) and where alignment occurs (across views vs. within views).
>
> $~$
>
>
> **Adaptive or learnable thresholding**
>
> Yes, we agree. This is exactly why we explicitly list it as a limitation of the current method. If we already acknowledge that an adaptive or learnable threshold could further improve region discovery, then isn't exploring such alternatives naturally part of future work rather than a requirement for validating the present contribution? And given that CRISP, even with this suboptimal and fixed threshold, still substantially outperforms state-of-the-art methods on dense tasks, does this limitation actually hinder the effectiveness or validity of the method as presented? Or does it instead highlight that there is clear potential for further gains beyond the strong results we already report?
>
> $~$
>
> **iBOT initialisation and finetuning performance**
>
> These concerns are addressed in detail in our responses to reviewers BJfD and 4Ecc; please refer to those discussions.

---

> > ### Author Response · Authors · 2025-11-14
> > **Continue Rebuttal**
> >
> > **Response to the Questions**
> >
> > 1. Please refer to Figure 6 in the ablation study, where we analyse the sensitivity of CRISP to beta. While we agree that adaptive or scheduled thresholding could be beneficial, we intentionally leave this to future work. Importantly, even with a fixed and non-optimal threshold, CRISP already shows strong performance improvements across dense tasks, which indicates that the method is robust to this design choice.
> >
> > 2. Due to resource constraints, we did not perform an extensive sweep over the lambda parameters. As stated in the paper, we used the simple setting lambda_i = 1 for all loss terms. But to respond to the reviewer question, we believe that increasing lambda_region value might slightly increase perforamnce on dense task, but affect the performance on global tasks.
> >
> > 3. Recent SSL approaches commonly follow a staged training scheme, models are first trained with a simpler objective, and once representations stabilise, additional losses are introduced (e.g., iBOT itself first trains with the DINO loss before adding the patch-level loss). If we were to train CRISP from scratch, we would follow this established recipe, and we expect performance to improve further. However, we do not have the computational resources required for such long training runs.
> > For fairness, we did continue training iBOT for an additional 200 epochs under its own objective, and its performance decreased, reinforcing that CRISP's improvements do not come from additional compute budget.
> >
> > 4. We only evaluated CRISP under off-the-shelf settings. We do not expect the performance gap to persist under full end-to-end fine-tuning. Fine-tuning introduces new factors, i.e. task-specific losses, decoder design, and optimisation recipes. Our focus is on reducing the gap between of-the-shelf and fine-tuning performance. We expect only a slight improvement in performance as task-specific training dominates.
> >
> > 5. The complexity of concept-token aggregation is similar to that of a standard transformer block. Since we already compute full attention for each batch, restricting aggregation to within-region tokens does not change the asymptotic complexity. When patch size decreases (e.g., 8x8), the number of tokens increases, and the computational cost scales correspondingly, just as it does for any transformer architecture operating at higher resolution.
> >
> > 6. The discovered regions serve primarily as a representation-learning mechanism, not as explicit segmentations. They do not need to perfectly align with object boundaries to be effective. In fact, regions are sometimes small, fragmented, or correspond only to parts of objects, yet these partial, view-consistent groupings are sufficient to yield the observed performance gains. Matching these semantically coherent local groups across augmentations encourages stronger region-level consistency than token-wise matching alone, which explains the improvement regardless of occasional boundary spillover.
> >
> > -------------------------
> >
> >
> > We thank the reviewer again for the insightful questions, which have helped clarify our contribution.

---

### Author Response · Authors · 2025-11-14
**Our Work Was Not Fairly Evaluated – Final Statement Before Withdrawal**

We are withdrawing this submission due to deadline for another venue. Before doing so, we felt it necessary to express our deep frustration with how our work was evaluated. The current reviews give extremely low scores without engaging in any meaningful way with the methodology, the technical contribution, or the core idea of the paper. Instead, the focus was almost entirely on a small set of requested experiments, some of which are not logically grounded and demonstrate a misunderstanding of the existing literature.

For example, several reviewers asked us to use DINOv2 for initialisation. This request itself reveals a misunderstanding of the role of initialisation in our setting, it is not an ablation factor here, and we were simply being transparent by stating that we initialise directly from iBOT rather than retraining objectives in staged pipelines, as many recent SSL works do. **More importantly**, the request is fundamentally flawed. DINOv2 follows the same design principles as iBOT, with the main differences arising from hyper-parameter optimisation, large-scale "in-house" pretraining and distillation from much larger models. Asking us to replace our initialisation with DINOv2 and "see if performance increase" ignores these underlying differences and makes the proposed experiment meaningless, and in practice, completely infeasible, even if the compute were available.

It is extremely disheartening to receive low scores based solely on such requests, without any substantive discussion of the method itself, its novelty, simplicity, or its strong empirical impact on dense prediction tasks while maintaining global performance. As authors, we rely on reviewers to assess the ideas, not only to list additional experiments.

We recognise that reviewing is voluntary and time-consuming. However, the current process, as reflected in our reviews, did not provide a fair or meaningful assessment of our work. It is disappointing that the evaluation seems to have been driven by surface-level concerns rather than a genuine attempt to understand the contribution.

In the remaining short time before withdrawal, we will still do our best to address the other reviewers’ comments, even though there is realistically no time for meaningful re-engagement from their side.

Given these circumstances and the constraints of the upcoming deadline, we are withdrawing the submission. We hope that future review processes, for us and for others, will better reflect the merits of the work rather than rely on incomplete or misaligned criteria.

---

### Note · Authors · 2025-11-14

**Comment:**

We believe that the evaluation does not fairly reflect the contribution of the proposed work.

**Withdrawal Confirmation:**

I have read and agree with the venue's withdrawal policy on behalf of myself and my co-authors.